# Innovative Thinking, Infinite Humor: Humor Research of Large Language Models through Structured Thought Leaps

**Han Wang**[*1,2], **Yilin Zhao**[*2], **Dian Li**[†‡2], **Xiaohan Wang**[1,2], **Gang Liu**[2], **Xuguang Lan**[† 1], **Hui Wang**[2]

Xi'an Jiaotong University[1]; Tencent QQ[2]
```
reload7@stu.xjtu.edu.cn,
{yilinnzhao,goodli,shawnbywang,sinbadliu}@tencent.com,
xglan@mail.xjtu.edu.cn, joltwang@tencent.com
```

## Abstract

Humor is previously regarded as a gift exclusive to humans for the following reasons. Humor is a culturally nuanced aspect of human language, presenting challenges for its understanding and generation. Humor generation necessitates a multi-hop reasoning process, with each hop founded on proper rationales. Although many studies, such as those related to GPT-o1, focus on logical reasoning with reflection and correction, they still fall short in humor generation. Due to the sparsity of the knowledge graph in creative thinking, it is arduous to achieve multi-hop reasoning. Consequently, in this paper, we propose a more robust framework for addressing the humor reasoning task, named LoL. LoL aims to inject external information to mitigate the sparsity of the knowledge graph, thereby enabling multi-hop reasoning. In the first stage of LoL, we put forward an automatic instruction-evolution method to incorporate the deeper and broader thinking processes underlying humor. Judgment-oriented instructions are devised to enhance the model's judgment capability, dynamically supplementing and updating the sparse knowledge graph. Subsequently, through reinforcement learning, the reasoning logic for each online-generated response is extracted using GPT-4o. In this process, external knowledge is re-introduced to aid the model in logical reasoning and the learning of human preferences. Finally, experimental results indicate that the combination of these two processes can enhance both the model's judgment ability and its generative capacity. These findings deepen our comprehension of the creative capabilities of large language models (LLMs) and offer approaches to boost LLMs' creative abilities for cross-domain innovative applications.

## 1 Introduction

Currently, humor is attractive because, as shown in Figure 1, it requires a burst of inspiration, which is still difficult for humans. The difficulty lies in the multi-hop reasoning process that fosters creativity. Each hop in this process is based on proper rationales. Without an understanding of these rationales, it is difficult for the model to grasp the internal humorous logic, making it prone to relying on pattern recognition.

So far, the Creative Leap of Thought (CLoT) paradigm (Xu et al., 2024) has developed two basic abilities to facilitate humor generation: selection skill and ranking skill. With these two basic skills, CLoT introduces nouns as instruction back-translation for self-improvement. However, only question-answer pairs, i.e., the beginning and ending of the multi-hop reasoning path, are utilized to train the model. As mentioned in CLoT, this process only captures the inherent creative patterns within the data which impairs the generalization ability and fails to stimulate "thinking outside the

---

[*]Equal Contribution. Work done during internship at Tencent QQ, as a part of QQ MLLM project
[†]corresponding author
[‡]Project leader of QQ MLLM project

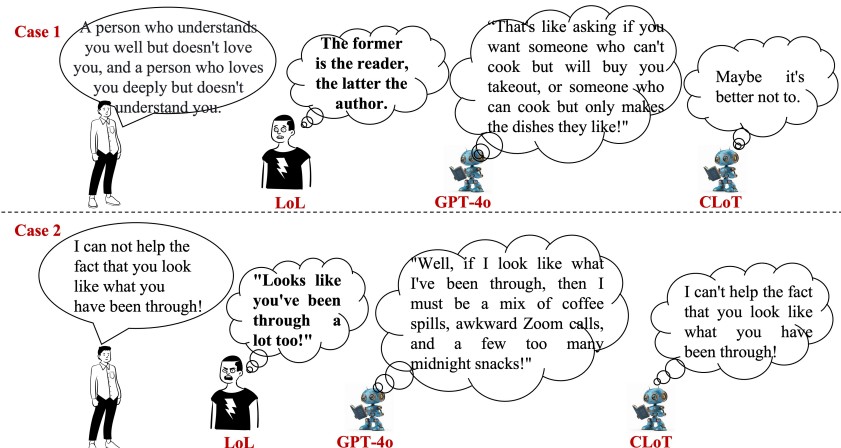

Figure 1: English comparison showcase (more showcases are in Appendix A.5). Compared to GPT-4o and CLoT, LoL provides shorter and more conversational answers to questions. For instance in Case 2, while LoL and CLoT may convey the same meaning, their different expressions produce different effects. Brief responses leave room for readers to ponder, enhancing interest and interactivity.

box" for generating novel ideas. Large language models (LLMs) such as GPT-4o or o1 (Lightman et al., 2023) and QwQ (Team, 2024b), which show superior performance in almost all reasoning tasks, do not perform exceptionally well in humor generation, as shown in Figure 1. Moreover, humor-related works (Xu, 2024; Xu et al., 2024) focus on a specific aspect of humor, like puns or proverbs, while humor also encompasses elements such as irony, limiting the range in real-world applications.

For the multi-hop humor reasoning problem, understanding is fundamental for endowing LLMs with reasoning ability to avoid getting trapped in memorizing patterns. The introduction and augmentation of external knowledge help LLMs understand the underlying logic and rationale. Additionally, a reward model can optimize the behavior of large language models by providing feedback, enabling them to produce outputs that better meet expectations. Due to the subjectivity of humor, a unified score may contain significant noise. Thus, the judgment skill is essential for providing feedback to further enhance LLMs' reasoning ability. Finally, with these two basic skills, humor understanding and judgment, the ability to generate humor can be improved.

Therefore, we propose a more robust framework named LoL to address the humor reasoning task that current LLMs find challenging. LoL consists of two-stage training: the supervised Fine-Tuning (SFT) stage and the Direct Preference Optimization (DPO) stage. In the first stage, we develop human-designed judgment-related instructions and their derivatives to train the model's humor judgment capabilities. Additionally, we propose an automatic instruction expansion method for humorous conversations to inject and augment knowledge into the original training data, mimicking the human thinking process step by step. This will help LLMs deepen and broaden their understanding of humor content. In the second stage, the reasoning rationale for each online-generated response is extracted using GPT-4o. In this process, external knowledge is introduced again to assist the model in logical reasoning and learning human preferences. The judgment capability from the first stage can also be useful for expanding the preference-pair dataset and further supplementing rationales.

We evaluated the humor judgment abilities of various large language models on both Chinese and English humor datasets. Experiments demonstrate that LoL outperforms other models on almost all test sets. Additional confirmatory experiments were conducted to show that LoL enhances the model's divergent thinking ability and effectiveness in humor generation. Our contributions are summarized as follows.

1. We propose an automatic instruction-evolution system for conversation data. A three-agent system is introduced to inject and augment knowledge into the original training data. This

will facilitate LLMs in deepening and broadening their understanding of the underlying logic and rationales.

2. We propose a teacher-student prompt system to enhance the judgment ability of LLMs. Through the automatic construction of conversation data between the teacher and the student, LLMs learn the teacher's judgment of the student's thinking.

3. Experimental results demonstrate that we can enhance both the model's judgment and generative capabilities and achieve state-of-the-art performance.

## 2 METHOD

### 2.1 PROBLEM FORMULATION

The multi-hop reasoning issue can be framed as a knowledge-graph (KG) problem. In this context, nodes (or concepts) along the multi-hop path are entities within the KG, and the rationales for enabling multi-hop reasoning are relations within the KG. By enriching the entities and relations in the KG, we can more easily explore the self-evolved path in multi-hop reasoning.

In general, the knowledge graph $\mathcal{G}$ is defined as a set of triples $\mathcal{G} = \{(e, r, e') \mid e, e' \in \mathcal{E}, \ r \in \mathcal{R}\}$, where $\mathcal{E}$ is the set of entities and $\mathcal{R}$ is the set of relations. Each triple represents a relation $r$ from the head entity $e$ to the tail entity $e'$ (Sun et al., 2023a; Yang et al., 2023). In the specific application of humor generation, we consider a knowledge graph composed of question-related entities $\mathcal{E}_Q$ and answer-related entities $\mathcal{E}_A$. The intersection $\mathcal{E}_Z = \mathcal{E}_Q \cap \mathcal{E}_A$ is regarded as the set of correlation entities (we refer to them as correlation entities here). Given the creative and unexpected nature of humor, as well as the existence of causal relationships between questions and answers, $\mathcal{E}_Z$ may consist of pseudo-correlation entities between $\mathcal{E}_Q$ and $\mathcal{E}_A$, such as those involved in puns (i.e., $\mathcal{E}_Z \rightarrow \mathcal{E}_Q$, $\mathcal{E}_Z \rightarrow \mathcal{E}_A$, and $\mathcal{E}_Q \rightarrow \mathcal{E}_A$). Additionally, it can also follow the pattern $\mathcal{E}_Q \rightarrow \mathcal{E}_Z \rightarrow \mathcal{E}_A$, as shown in Figure 17. Clearly, it can be inferred that $\mathcal{E}_Z$ is pivotal for enabling the multi-hop in the humor reasoning.

Therefore, we formulate the causal relation $R_c$ into a verbal description as shown in Figure 12, which contains the correlation entities $\mathcal{E}_Z$ either explicitly or implicitly. Finally, our objective is to expand the scopes of $\mathcal{E}_Q$ and $\mathcal{E}_A$, and utilize the causal relationship to structure the reasoning path. This is beneficial for mitigating the information insufficiency problem and further enabling humor reasoning.

The overall training framework is illustrated in Figure 4. In the first stage, supervised fine-tuning (SFT), we randomly initialize a LoRA model and train it using both single-turn and multi-turn question-answer format data. In the second stage, Direct Preference Learning (DPO), the model from the first stage serves as a reference model and is frozen to act as the judgment model when self-evolving. The tunable model is trained with preference question-answer data, which helps the model improve its logical reasoning and learn human preferences.

### 2.2 DIVERSE INSTRUCTION EXPANSION AND TUNING

A reward model can optimize the behavior of large language models by providing feedback, enabling them to generate outputs that better meet expectations. However, due to the subjectivity inherent in humor, a unified scoring system (i.e., a pointwise reward model) may be plagued by significant noise, such as topics, background and so on. Additionally, human-voting (i.e., Likes) data from the community is readily available. Transforming it into pairwise data is not only easier but also more in line with human intuition of selecting the more humorous one from two responses. Consequently, we adopt a judgement model (i.e., pairwise reward model) to offer feedback, thereby further enhancing the reasoning capabilities of large language models (LLMs).

To stimulate the judgment capability of a model, we manually design a judgment-oriented template as shown in Figure 2. Additionally, the understanding capability is enhanced to improve both the judgment and generation capabilities through automatic instruction evolution.

**Judgement Template Design.** Question-answer data with human-voting annotations are utilized for judgment tasks.

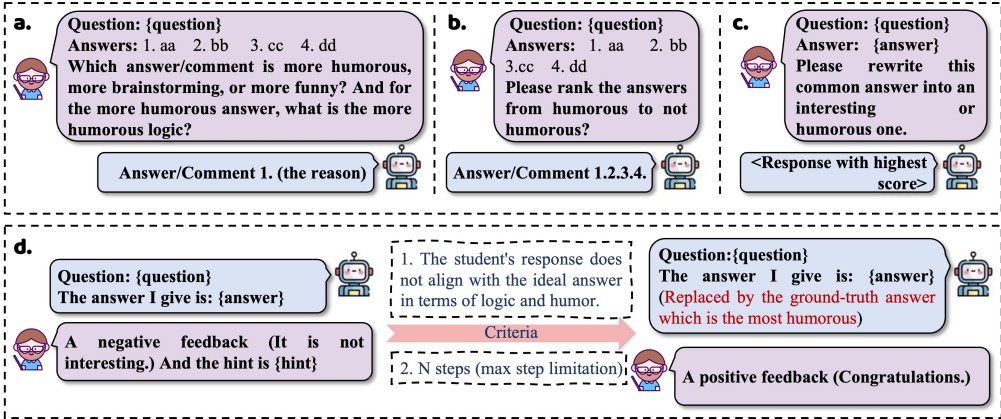

Figure 2: The details of judgement-oriented instructions templates.

(1) *Selection and Ranking Skill.* Two basic templates are designed, as shown in Figure 2 (a) and (b). Single-choice and ranking questions are basic tasks for developing judgment capabilities. Furthermore, to deepen the model's understanding of the contrast between "humorous" and "non-humorous", two additional tasks are proposed as shown in Figure 2 (c) and (d).

(2) *Answer Rewriting* (template in Figure 2(c)). We identify the answer receiving the most human votes as the most humorous one. Subsequently, GPT-4o is employed to rewrite this most humorous answer into a non-humorous version. Finally, the model is trained to transform a non-humorous statement into a humorous expression using single-round session data format.

(3) *Teacher-Student Prompt Loop* (template in Figure 2(d)). Similar to the human thought process, the complete exploration process encompasses trial-and-error, reflection, and backtracking. Within this process, the ability to make sound judgments in a long chain of thought is crucial. Therefore, a guided conversation process between two agents is proposed to enhance the reflection, and backtracking. We utilize two GPT-4o models and designate one as the teacher and the other as the student. The teacher provides a judgment conclusion and provides prompts based on the given question and the ideal answer. The student generates an answer using the teacher's prompts and the original question. Two criteria as shown in Figure 2(d) are applied to stop this multi-turn conversation.

**Automatic Instruction Expansion.** In the human cognitive process, mechanisms such as trial and error, reflection, and backtracking are inherently supported by a robust understanding of concepts. To emulate the deep and extensive understanding characteristic of human cognition, we propose a three-agent system inspired by automatic instruction evolution, which leverages the extensive world knowledge encapsulated in large language models (LLMs) to autonomously enrich prompts within the realm of humor comprehension. This methodology not only facilitates the exploration of unconventional associations among disparate concepts but also enhances the model's ability to establish and strengthen interconnections between diverse ideas. The process is shown in the Figure 3.

Given a seed conversation dataset $D = \{(q_k^0, a_k^0)\}_{k=0}^N$ where $q_k^0$ denotes questions and $a_k^0$ represents humorous responses. Let $I_0 = \{I_k | I_k = (q_k^0, a_k^0, i_k^0)\}_{k=0}^N$ be the initial instruction set, where each instruction $i_k^0$ contextualizes the dialogue pair $(q_k^0, a_k^0)$. The framework implements a 3-round evolutionary process through three cooperative agents. The iterative process terminates when either Criterion-2 is satisfied or maximum evolution rounds ($t \geq 3$ in our setting) are reached. Each iteration produces enhanced instruction-output pairs $(I_{new}, y_{new})$ where $y_{new} \in Y$ denotes optimized responses under the augmented instruction space.

*Generation*: The Generators module streamlines the process of instruction generation, enabling the automated generation of instruction data based on a seed conversation. Unlike evol-instruct methods in math and other fields, we design an "imaginator" in the generation process for humor reasoning. It tries to guess the seed conversation based on the answer from evolutionary instruction. This can cause a shift in the storyline within the conversation topic. In other words, it tries to explore the boundaries of the knowledge graph under the context of the seed conversation and enhance creativity.

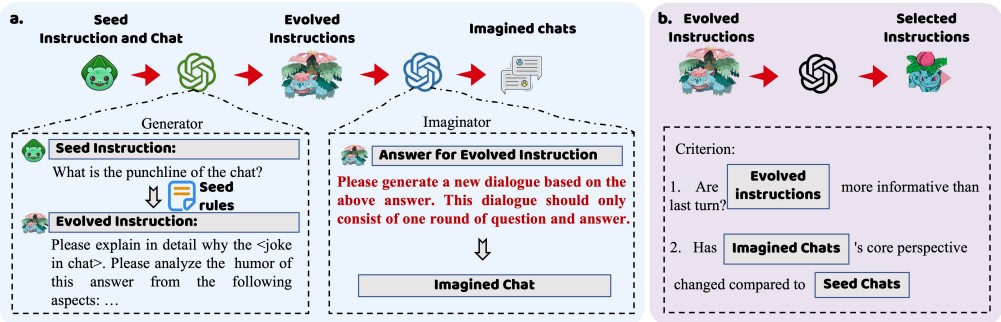

Figure 3: The detail of AIE. And the detailed process is shown in Algorithm 1

*Selection*: The Selectors module is crafted to streamline the instruction-filtering process, enabling the curation of instruction datasets from evolved instruction data. In contrast to evol-instruct techniques in mathematics and other domains, this approach serves not merely as a means to unleash the capabilities of LLMs, but also to boost the capacity for thinking outside the box, thereby further actualizing creativity. Therefore, we design a critic related to the conversation topic, which limits the model's thinking from being overly divergent.

Finally, the system outputs multi-turn question-answer data which format is $\{(q_k^0, a_k^0), (I_k^0, y_k^0), \ldots, (q_k^{m_k}, a_k^{m_k}), (I_k^{m_k}, y_k^{m_k})\}_{k=0}^N$, where $m_k$ is the maximum number of communication rounds between the three agents. Finally, all above human-designed and automatic expanding data are involved in training.

## 2.3 GUIDE EXPLORATION AND SELF-IMPROVEMENT TUNING

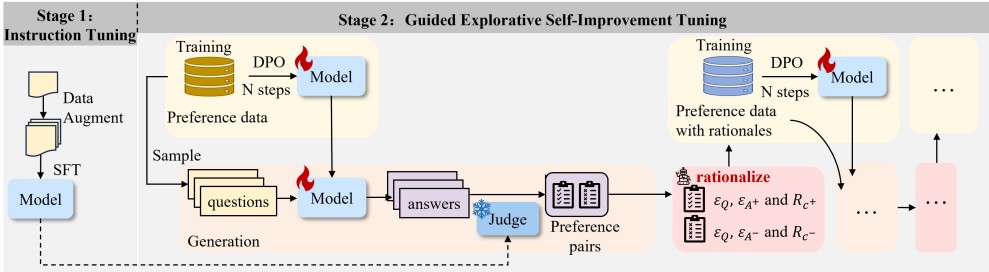

Figure 4: Pipeline of training process.

In the previous section, we developed a model with judgment and understanding capabilities. Through multiple rounds of dialogue, we strengthened the relationships among diversified entities $\mathcal{E}_Q$ and $\mathcal{E}_A$. At this stage, the policy model enhances its performance through online self-learning, using a dataset of (question, humorous answer, non-humorous answer) tuples: $(q, a^+, a^-)$. Specifically, as shown in Figure 4, it learns by repeatedly sampling responses, evaluating answer correctness with the capability from Section 2.2, and updating its parameters online using Direct Preference Optimization (Rafailov et al., 2024).

As detailed in Algorithm 2, we start with an initial offline preference dataset without rationales, $D_0 = \{(q_i, a_i^+, a_i^-)\}_{i=0}^M$, where $M$ is the number of training data. The model trained in Section 2.2 is copied into two. One, denoted as $\pi^*$, is frozen to judge whether sampled answers are humorous for online data augmentation. The other, $\pi$, is trained to have a more robust generative ability.

During training, the policy model $\pi$ trains on $D_0$ for a few steps. Every $T$ steps, $\pi$ selects $l$ questions from $D_0$ to sample two answers or each question. The judge model $\pi^*$ then determines the more humorous one, constructing new pairwise data incorporated into the training dataset. Additionally, a causal inference expert (GPT-4o) provides rationale for each answer relative to the question (see

Appendix A.4 for details). This process expands the preference dataset to $D = D_0 \cup \widetilde{D}$, where $\widetilde{D} = \{(q, \widetilde{a}^+, \widetilde{a}^-, \widetilde{r}^+, \widetilde{r}^-)\}$.

## 3 EXPERIMENTS

We collect humor-related data with human-voting from humor games and communities, and then organize it into the format mentioned in Section 2.2. Since we enhance the generation ability based on the judgment capability, it is crucial to verify the performance in humor-judgment tasks. Thus, we construct single-choice questions at different difficulty levels and carry out the consensus generation task validation.

### 3.1 DATASETS SOURCE

(1) Oogiri-GO (Zhong et al., 2024). In the "Oogiri game", participants are required to give unexpected and humorous responses to given images, text, or both. This game demands a sudden burst of insight and strong associative thinking within the given context. Similar to the processing method of CLoT (Zhong et al., 2024), we randomly select 95% of the samples to construct the training dataset, and the remaining 5% is used to form the test dataset for validation and analysis.

(2) SemEval 2021 Task 7 Meaney et al. (2021) contains binary labels and ratings collected from a balanced age group ranging from 18 to 70 years old. Data with binary labels and ratings are utilized to construct the easy-case task and the hard-case task mentioned in Section 3.2 respectively.

(3) SemEval 2020 Task 7 (Hossain et al., 2020a) is a game to change a word in a headline to make it funnier. It also contains binary labels and ratings which are utilized to construct the easy-case task and the hard-case task.

(4) Chinese Community Data. We collect data from various Chinese communities, including Ruozhiba and others. Human-voting (i.e., Likes) labels are readily available, and we use them to construct both easy-case and hard-case tasks.

### 3.2 TASKS CONSTRUCTION

(1) Inspired by the task design in CLoT(Zhong et al., 2024), we develop judgement-related tasks as follows.

**Easy-case Task (i.e. $2T1$).** Single-choice-from-two-options questions are constructed from the binary-label data mentioned above and the data with the largest gap in human-voting counts. The construction adheres to the template depicted in Figure 2(a).

**Hard-case Task (i.e. $2T1(Hard), 3T1, 4T1$).** Single-choice-from-two(or three or four)-options questions are constructed from the same source with a closer gap in human-voting counts. The construction adheres to the template depicted in Figure 2(a).

(2) Inspired by the generation tasks designed in CLoT(Zhong et al., 2024), the Divergent Associate Thinking (DAT) task (Olson et al., 2021) to validate the capability of creativity is carried out. Additionally, human evaluation will also be validated.

Table 1: Humor judgement validation on ruozhiba dataset

| Model | 2T1 |
|---|---|
| GPT-4o | 76.40 |
| QWEN1.5-32B | 68.85 |
| CLoT | 50.20 |
| QwQ | 90.80 |
| LoL | 95.35 |

### 3.3 RESULTS ANALYSIS

**Evaluation on Judgement tasks in English.** We validate the top-1 accuracy of completing each judgement task and show the performance of several models in Table 3. Overall, compared with open-source language models such as LLaMA3, QWEN and so on, state-of-the-art closed-source LLMs exhibit impressive zero-shot performance in humor judgement tasks. Model trained by LoL has significantly improved compared to other models (such as LLAMA3-70B and

GPT-4o). Specifically, the average accuracy in diverse English benchmarks has increased by 4.55% and 5.91% respectively.

**Evaluation on Judgement tasks in Chinese.** We also evaluate the accuracy rate (acc%) of completing each selection task in Chinese and show the performance of several models in Table 2. Overall, compared with open-source language models including LLaMA3 and QWEN, the state-of-the-art closed-source large language models show impressive zero-shot performance on humor judgement tasks in Chinese. The model trained by LoL also shows a significant improvement compared to other models (such as GPT-4o) (with an average accuracy in diverse Chinese benchmark increase of 16.22%).

We also conduct an additional experiment on the Ruozhiba dataset[*], which most well-known large language models (LLMs) have been trained on. We asked GPT-4o to rewrite the Ruozhiba queries into question-answer pairs, placing the punchline in the answer section. Then, we asked GPT-4o again to rewrite the ground-truth answers into non - humorous versions. Based on the positive-negative pair data, the LLMs were tested, and the results are shown in Table 9. The results indicate that CLoST also achieves state-of-the-art performance on the Ruozhiba dataset.

Table 2: The accuracy (%) of choice questions on various Algorithms in Chinese benchmarks.

| Model | | Chinese Benchmark | |
|---|---|---|---|
| | | 2T1 | 2T1(hard) |
| GPT | 4o | 64.98 | 63.49 |
| LLAMA3 | 8B | 50.72 | 57.44 |
| | 70B | 59.48 | 61.22 |
| QWEN1.5 | 7B | 54.82 | 51.71 |
| | 14B | 53.45 | 57.41 |
| | 32B | 52.71 | 56.27 |
| QWEN2 | 7B | 51.99 | 58.17 |
| | 57B | 65.91 | 57.03 |
| QWEN2.5 | 32B | 61.53 | 60.46 |
| Baichuan2 | 13B | 50.56 | 53.61 |
| CLoT | 7B | 52.12 | 34.46 |
| QwQ | 32B | 59.75 | 57.04 |
| QwQ+LoL | 32B | 88.91 | 63.12 |
| OURS | 32B | **90.95** | **69.97** |

**Evaluation on Creative-thinking validation in Generation Tasks.** To evaluate the associative generalization capability of LoL, we test it on a creative task known as the Divergent Association Task (DAT). The DAT is a classic creativity test in which testee write 10 unrelated words, and words with greater "distances" in their context embeddings indicate more divergent thinking. In the Chinese creativity test, we utilize Chinese Word Vectors (Li et al., 2018) to calculate the DAT score. First, we provide specific words and ask the model to generate associations and imaginations, obtaining 10 associated words. Then, we use these 11 words to calculate the DAT score. Additionally, we test how the model's DAT score varies as the number of test words increases. It can be observed that as the number of words increases, the DAT score also tends to stabilize. From the average of each domain, it can be seen that LoL has the highest score after stabilization. As shown in Figure 5, LoL has a slight performance improvement in the mean value of DAT compared to Qwen1.5-32B, GPT-4o and CLoT.

In addition, we employ T-SNE to project the embedding vectors of these words into a two-dimensional space. specifically, the target word is positioned at the center, and a circle is drawn with a radius equal to the Euclidean distance between the target word's embedding and that of the farthest associated word. In Figure 5(c) and (d), the embedding vectors of five target words and their associated words are illustrated, with different colors representing different target vectors. A larger circle indicates a broader semantic range for the target word, implying a greater number of associated words. In both tests, LoL outperforms previous works including SOTA models like GPT-4o.

**Human Evaluation in Generation Tasks.** We conduct a user-preference study to test the creativity and humor of LLMs. Here, we select four LLMs (LoL, GPT-4o, QWEN1.5-32B, CLoT) to generate responses for a total of 200 text-based questions. We present a question and several corresponding replies from the four LLMs, and ask users to choose the most creative and humorous response. Figure 6(c) summarizes the statistical analysis of 3000 valid surveys. Figure 6(b) shows the win-rate calculated based on the problem dimension. The results indicate that users have a strong inclination

---

[*]https://github.com/Leymore/ruozhiba/tree/main?tab=readme-ov-file

TABLE 3: THE ACCURACY (%) OF CHOICE QUESTIONS ON VARIOUS ALGORITHMS IN ENGLISH BENCHMARKS.

| MODEL | | SEMEVAL 2021 | | | | SEMEVAL 2020 | OOGIRI-GO |
|---|---|---|---|---|---|---|---|
| | | 2T1 | 2T1(HARD) | 3T1 | 4T1 | 2T1 | 2T1 |
| GPT | 4O | 85.09 | **60.77** | 43.71 | 34.63 | 55.08 | 85.09 |
| LLAMA3 | 8B | 43.85 | 54.23 | 39.81 | 29.57 | 59.93 | 72.05 |
| | 70B | 93.60 | 58.08 | 39.81 | 31.82 | 60.73 | 88.51 |
| QWEN1.5 | 7B | 62.04 | 52.02 | 31.54 | 24.89 | 50.46 | 36.65 |
| | 14B | 82.05 | 51.04 | 30.92 | 24.24 | 50.38 | 53.73 |
| | 32B | 68.01 | 52.57 | 35.38 | 28.79 | 56.39 | 68.01 |
| QWEN2 | 7B | 56.55 | 50.63 | 32.31 | 23.38 | 50.08 | 62.11 |
| | 57B | 83.30 | 52.02 | 37.08 | 28.79 | 48.29 | 48.14 |
| QWEN2.5 | 32B | 94.00 | 55.22 | 34.77 | 27.92 | 58.71 | 81.68 |
| BAICHUAN2 | 13B | 51.70 | 52.71 | 35.69 | 24.24 | 51.45 | 50.00 |
| CLoT | 7B | 52.49 | 51.74 | 34.46 | 23.59 | 53.50 | 52.49 |
| QwQ | 32B | 80.05 | 53.06 | 33.49 | 24.66 | 56.58 | 59.63 |
| OURS | 32B | **96.58** | 57.45 | **48.06** | **35.90** | **64.57** | **97.20** |

TABLE 4: ABLATION ON ENGLISH BENCHMARKS.

| MODEL | SEMEVAL 2021 | | | | OOGIRI-GO-EN |
|---|---|---|---|---|---|
| | 2T1 | 2T1(HARD) | 3T1 | 4T1 | 2T1 |
| QWEN1.5-32B | 68.01 | 52.57 | 35.38 | 28.79 | 68.01 |
| OOGIRI-GO+FIGURE 2(C) | 55.12 | 53.27 | 33.54 | 27.49 | 81.68 |
| OOGIRI-GO+FIGURE 2(C,D) | 89.70 | 50.28 | 30.46 | 17.32 | 95.96 |
| OOGIRI-GO+AIE | 88.40 | 50.07 | 30.92 | 19.48 | 96.58 |
| ALL+FIGURE 2(C,D) | 92.25 | 52.81 | 37.46 | 30.26 | 96.27 |
| ALL+AIE(DIET) | **94.25** | **53.10** | **45.46** | **32.26** | **97.20** |

to select the results of LoL, highlighting the high - quality creative content generated by CLoT. For more details of the user study, refer to the Appendix 1.

**Ablation Study**

We examine the ablation effects of different components in LoL and report the performance results in Table 4 and Table 5. In Table 4, line 1 presents the performance of QWEN-1.5-32B. Lines 2-4 show the performance of gradually adding methods on the Oogiri-GO-en dataset. The results indicate that with the increase in the number of tasks, especially in the teacher-student system, the judgment performance improves a lot. Employing AIE on the Oogiri-GO-en dataset only causes a slight decrease in performance. This might be caused by overfitting to the divergence of thought in the

Table 5: Ablation on Chinese benchmarks.

| MODEL | CHINESE BENCHMARK | |
|---|---|---|
| | 2T1 | 2T1(HARD) |
| QWEN1.5-32B | 52.71 | 56.27 |
| +FIGURE 2(C) | 63.56 | 63.88 |
| +FIGURE 2(C)(D) | 83.62 | **64.64** |
| +AIE (DIET) | **90.34** | **64.64** |

Oogiri-GO-en dataset. Lines 5-6 show that when all datasets with the introductory-part method are used to train the model, a further increase in performance is realized.

To evaluate the performance of GESIT, which primarily enhances the model's causal and logical reasoning capabilities (i.e., the ability to relate to internal contexts), we enlisted three experts to assess the logical coherence of responses on 200 examples from GPT-4o, GPT-4 and QWEN2-57B,

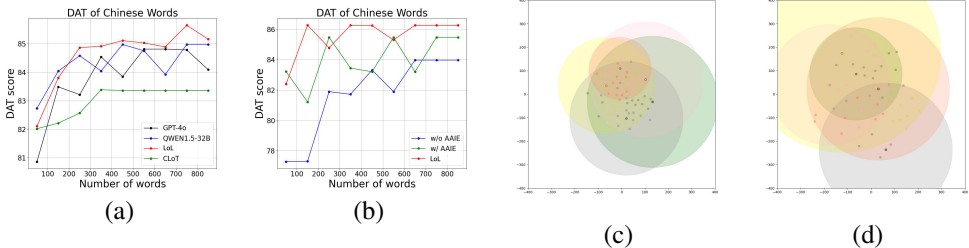

Figure 5: Divergent associate thinking (DAT) validate. (a). DAT score compared among LoL and three baselines (b). DAT score compared among different component of LoL (c). TSNE Results of Word Vectors Obtained by QWEN1.5-32B Associating Five Target Words. (d). TSNE Results of Word Vectors Obtained by LoL Associating Five Target Words.

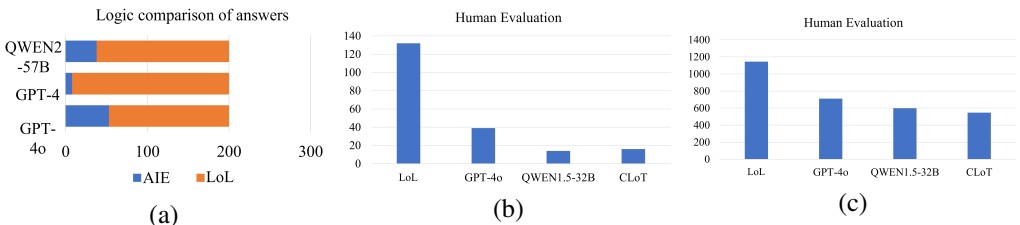

Figure 6: (a) GPT-4 and GPT-4o logically evaluates the output of the model after DIET and LoL respectively. (b). Human evaluation about the win rate statistics based on the problem dimension. (c). Human evaluation about the win rate based on the total number of votes received by the four LLMs.

respectively. The experimental results shown in Figure 6(a) demonstrate that incorporating GESIT significantly strengthens the logical reasoning in the model's answers. In addition, the DAT test is conducted on ablation study in Figure 5(b), which shows that AIE enhance the divergent associate thinking ability.

## 4 RELATED WORKS

**Large Language Models (LLMs) and Their Creativity.** Recently, language models (Bai et al., 2023; Wang et al., 2023; Liu et al., 2024; Chen et al., 2023) have garnered widespread attention due to their impressive reasoning capabilities (Wang et al., 2023; Saparov & He, 2022; Zeng et al., 2022; Driess et al., 2023; Dong et al., 2023; Ye et al., 2023; Liang et al., 2024). Additionally, an increasing number of studies are focusing on exploring the creativity of LLMs (Ling et al., 2023; Summers-Stay et al., 2023; Sun et al., 2023b; Bhavya et al., 2023), with applications in fields such as scientific discovery (Park et al., 2023; Kang et al., 2022; Hope et al., 2022; Liang et al., 2022; Huang et al., 2023) and creative writing (Swanson et al., 2021; Chakrabarty et al., 2022; Wu et al., 2022; Mirowski et al., 2023; Dang et al., 2023).

**Computational humor** is a branch of computational linguistics and artificial intelligence that utilizes computers to study humor (Binsted et al., 2006). It encompasses various tasks, including humor discrimination (Shahaf et al., 2015; Tanaka et al., 2022; Xu et al., 2022; Chen & Zhang, 2022; Kumar et al., 2022; Wu et al., 2021; Ofer & Shahaf, 2022; Xie et al., 2023; Meaney et al., 2021; Hossain et al., 2020a), humor interpretation (Hwang & Shwartz, 2023; Evans et al., 2019; Vásquez & Aslan, 2021), and humor generation (Amin & Burghardt, 2020; Zhang et al., 2020; Hossain et al., 2020b; Valitutti et al., 2013; Chaudhary et al., 2021). With the advancements in generative LLMs, humor generation has become a focal point due to its demand for creative thinking. (Zhong et al., 2024) extends the chain-of-thought paradigm into humor generation. However, only question-answer pairs and nouns as bask-translation, i.e., the beginning and ending of the multi-hop reasoning path, are utilized to train the model, and this process only captures the inherent creative patterns within the data which impairs the generalization ability and fails to inspire "thinking outside the box" for gen-

erating novel ideas. Therefore, we develop a reasoning process featuring the processes of thinking divergence and reflection for humor generation.

**Instruction evolutionary.** Recent attention has focused on the complex instruction-following capabilities of Large Language Models (LLMs), leading to the development of new evaluation benchmarks (Zhou et al., 2023; Jiang et al., 2023; Qin et al., 2024b). These studies consistently reveal that open-source LLMs lag behind proprietary models in their ability to follow complex instructions. However, there has been limited research on techniques to enhance this capability. Notable exceptions include Evol-Instruct (Xu et al., 2023; Zeng et al., 2024) and Conifer (Sun et al., 2024), which encourage LLMs to evolve instruction complexity and apply supervised fine-tuning (SFT) on the data generated from this process. Automated generation is a widely-used method for Evol-Instruct, minimizing the reliance on extensive human annotation or manual data collection. This approach utilizes LLMs to create large volumes of instructional data sourced from chat data (Chiang et al., 2023) or by expanding a small set of seed instructions (Wang et al., 2022; Xu et al., 2023; Li et al., 2023). The generated instructions are then used to derive corresponding inputs and outputs. In this paper, LoL develops an automatic instruction evolution method with three agents to strengthen the ability of thinking divergence and humor understanding.

**Large language models reasoning.** Human cognition involves two distinct modes of processing: one that is fast and intuitive, and the other that is deliberate and analytical. Currently, LLMs can not only generate rapid responses using learned patterns, but more significantly, simulate complex reasoning processes through mechanisms like chain-of-thought (Wu et al., 2022; Wei et al., 2022; Zhang et al., 2022; Yao et al., 2024; Long, 2023) or other forms of search, similar to how humans engage in deeper, step-by-step thinking (Fu et al., 2022; Lightman et al., 2023; Lai et al., 2024). However, these approaches build on existing LLMs without truly embedding the chain-of-thought ability within the models themselves. As a result, LLMs cannot inherently learn this reasoning capability, leading to active research on how to integrate it directly into model training. Most of these efforts(Qin et al., 2024a; Luong et al., 2024; Team, 2024b; Yang et al., 2024; Team, 2024a; Wang et al., 2024; o1 Team, 2024) focus on improving LLM reasoning by integrating process supervision, reinforcement learning (RL), and inference-time computation strategies such as guided search. By doing so, it shifts the focus from merely scaling model parameters during pre-training to leveraging smarter inference strategies at test time. These techniques help the model refine its reasoning step by step, allowing it to pause, evaluate intermediate reasoning, and select better solution pathways during test-time computation. All of these are adept at factual reasoning like math or coding reasoning, while humor reasoning is mostly non-factual reasoning requiring creative thinking.

## 5 CONCLUSION

In this paper, we introduce the LoL method aimed at enhancing the generation capabilities of large language models (LLMs). LoL commences by transforming humor datasets into instruction-tuning data to train LLMs, thus improving their logical thinking (LoT) and judgment abilities. Subsequently, LoL utilizes Guided Explorative Self-Improvement Tuning, enabling LLMs to generate more creative structured thought data by understanding rationales and to select high-quality data for self-refinement training. Experimental results illustrate the effectiveness and generalization ability of LoL across diverse tasks, such as witty response generation and humor discrimination.

## ACKNOWLEDGMENTS

This work was done during internship at Tencent QQ, as a part of QQ MLLM project and supported in part by NSFC under grant No.62125305, No.62088102, No. U23A20339, No. 62203348.

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

# A APPENDIX

## A.1 TRAINING AND EXPERIMENT DETAILS

**Training pipeline:**
LoL takes a two-stage training strategy. In the first process (supervised fine-tuning (SFT)), we randomly initialize a LoRA model. And we train the model with single-turn question-answer format data (data from Figure 2(a)(b)(c)) and muti-turn question-answer format data (data from Figure 2(d) and AIE). In the second process (Direct Preference Learning (DPO)), the first stage model serves as ref model, and it is frozen as judgement model. The tunable model is trained to improve the reasoning generation capability. At the beginning of stage 2, only preference question-answer data without rationale is fed into the tunable model for training. After several steps, the rationale for each online generated response is extracted using GPT-4o and the preference question-answer data with rationles are mixed into the original dataset. And in each batch, the ratio of 'w' and 'w/o' rationale is $1 : 1$.

**Implementation Details** We validate the validity of LoL mainly based on QWEN1.5-32B-Chat Bai et al. (2023) using the LoRA Hu et al. (2021) on 8 A100 GPUs. For the first stage, we train the model on 95% of the dataset mentioned above for 6 epochs using the AdamW optimizer with a learning rate of $3e - 4$. In the second stage, 5% of the dataset is used to train GESIT for 3 epochs using the AdamW optimizer with a learning rate of $2e - 4$. The models are tested on the tasks introduced in the previous part. And more parameters used in generation are listed in Table 7.

**Experiments parameters:**

Table 6: Parameters in Validate

| temperature | top k | top p | length penalty |
|---|---|---|---|
| 1 | 50 | 1.0 | 1.0 |

## A.2 DETAILS IN TEACHER-STUDENT LOOP IN FIGURE 2 (D)

Through the "teacher-student" interaction mechanism between two agents, the key characteristics of human thinking have been successfully simulated. The trial-and-error mechanism allows for the exploration of different solution paths; the reflection phase promotes an in-depth analysis of the root causes of errors; and the backtracking function enables the dynamic correction of the thinking path. This cognitive architecture makes the reasoning process of AI closer to the real-world problem-solving patterns of human experts. We utilize the data from the teacher agent to train the model with judgment ability. And an example of this loop is shown in Figure 7.

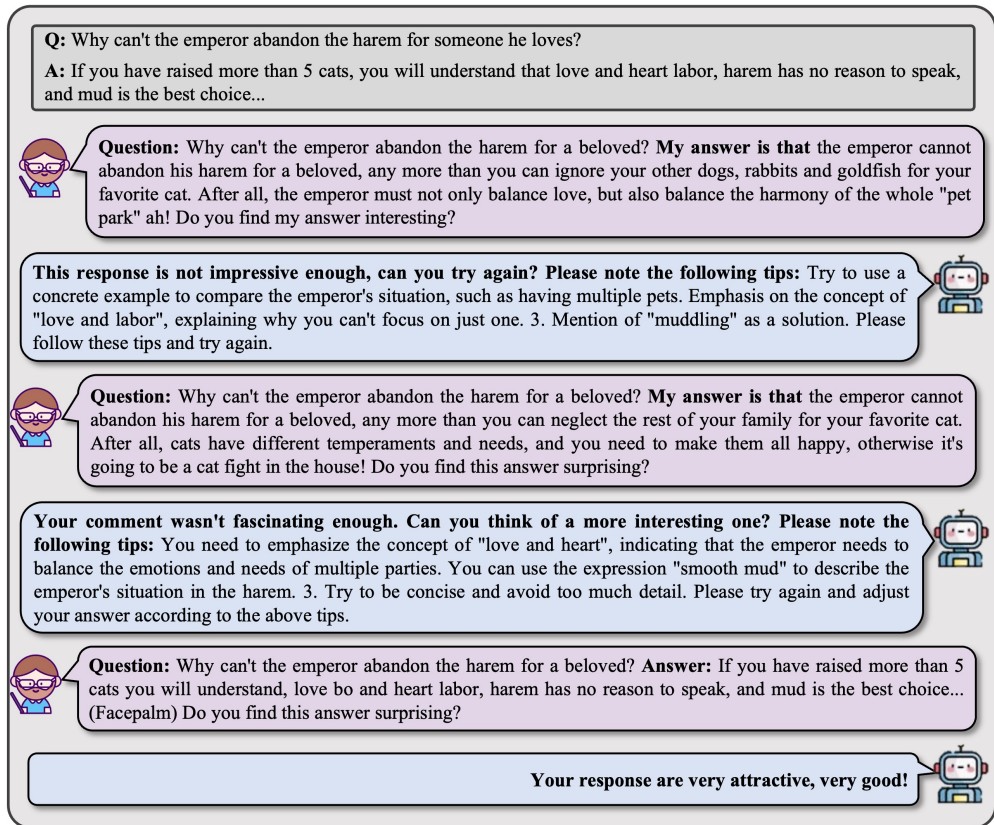

Figure 7: An example of *The teacher-student prompt loop* in Section 2.2

## A.3 DETAILS OF AIE

The detailed process of AIE is shown in Algorithm 1. The generator will inject more information into the conversation, and the imaginator will explore the boundaries of the knowledge graph behind the conversation. Finally, the criterion serves as a form of reject sampling for the augmented instructions.

An example of the generation process is shown in Figure 11. Before the evolution, the simple instruction will lead to very simple answers and understanding from the model. While after the evolution, the instruction can become more concrete and the answer will be more informative. And an example of the rule evolution is shown in Figure 18. Additionally, an example of rule evolution is shown in Figure 18. In addition, examples of the imagined conversations are shown in Figure 8 and Figure 9.

## A.4 DETAILS IN GESIT

GESIT develops an online DPO training frame with rationales to reinforce the reasoning ability. The example of the template is shown in Figure 18. And the training frame is shown in Algorithm 2.

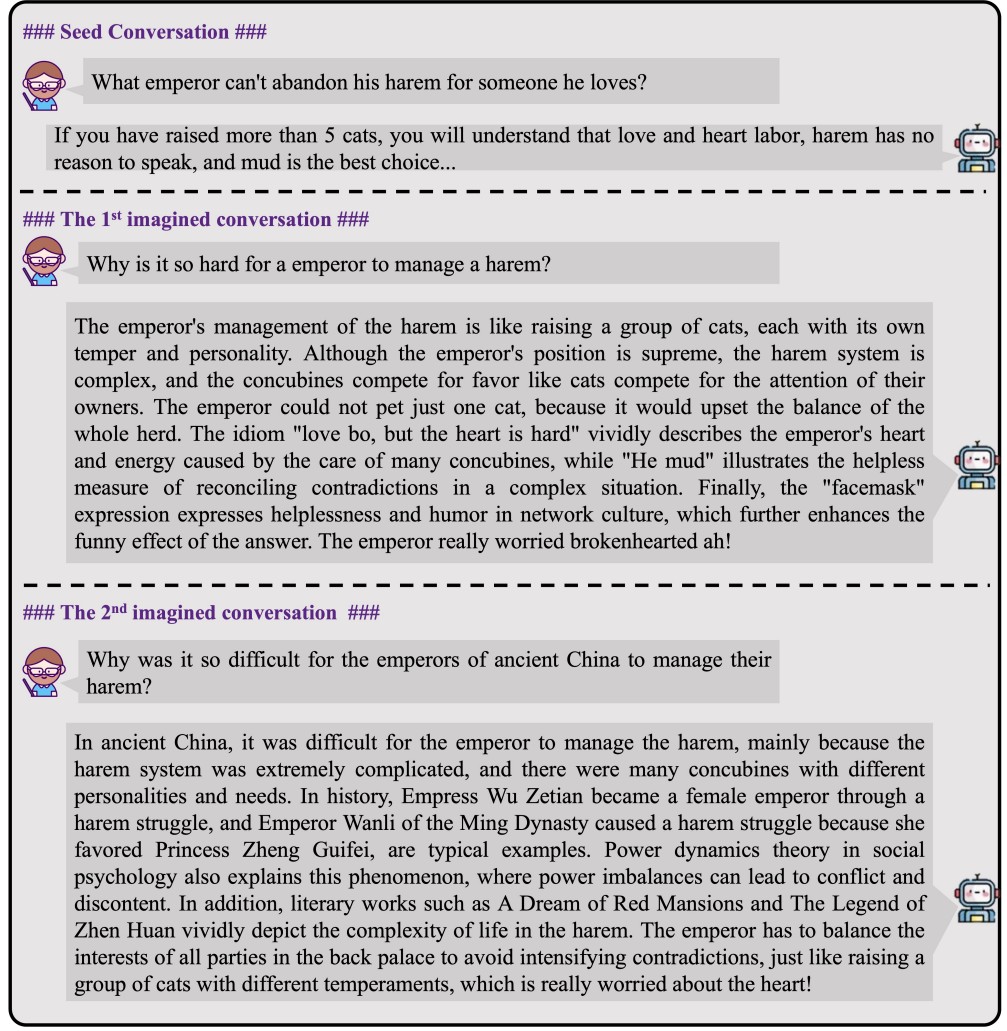

Figure 8: An example of the imagined process in AIE. It stops according to Criterion-1 in Figure 3(b).

## A.5 GENERATION SHOWCASE

The reasoning algorithm is presented in Algorithm 3. We pre-define some generation instructions, such as "Please generate a witty response to the question." We randomly select $n$ instructions, and the model generates $n$ different answers based on these instructions. The model's judgment ability assists in selecting the most humorous answer. The process and results indicate that the model's performance will not be restricted by a specific template.

**Successful showcase.** We randomly select posts from the internet and expect the model to provide the most humorous answer. Some examples are presented in Figure 15 and Figure 16. In summary, LoL's replies are usually more concise and direct. It is good at using rhetorical devices such as puns, onomatopoeia, repetition, and rhyme, and combining humor or irony with specific situations, which makes its replies more like "genius replies". In contrast, the responses of GPT-4o and CloT, while sometimes humorous, may be more verbose or lack an immediate punch.

To better demonstrate an example of the generation capability, we input the conversation into GPT-4o and let it imagine the details of the conversation, such as the scene, the characters, and especially their emotions. Finally, we input the results into Midjourney to generate an image based on the detailed description from GPT-4o. The examples are shown in Figure 13 and Figure 14.

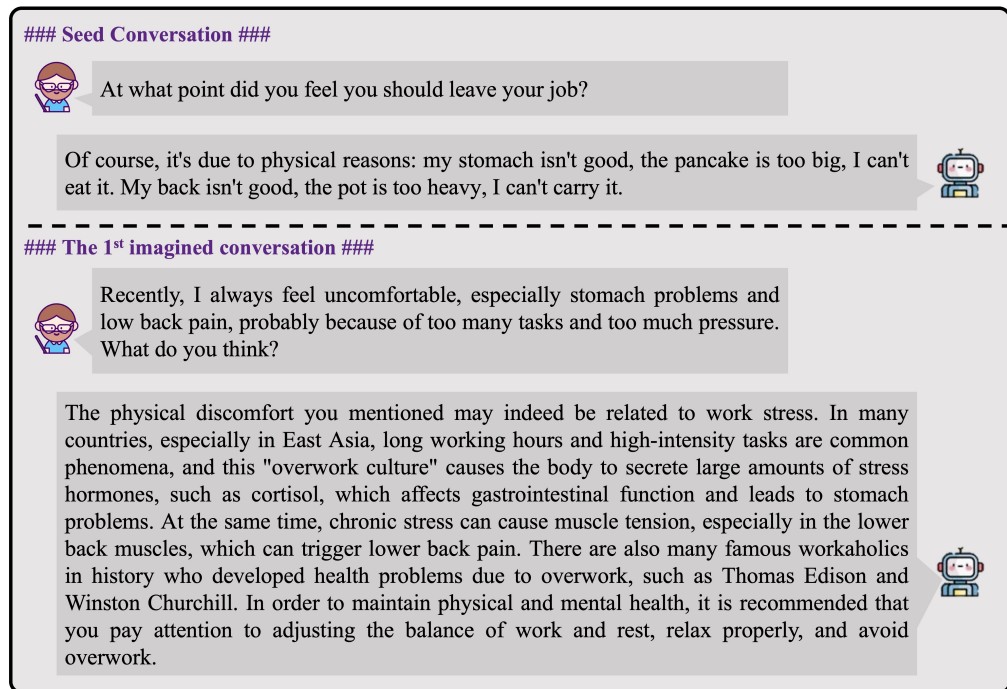

Figure 9: An example of the imagined process in AIE is presented here. It stops according to Criterion-2 in Figure 3(b), i.e., the core idea or joke of the imagined conversation has shifted from the seed conversation.

| 问题 | LoL | GPT-4o | CLoT |
|---|---|---|---|
| 谁会不喜欢小狗呢，大狗小狗都很爱 | 汪！ | 谁会不喜欢小狗呢？可能只有猫咪吧，它们可是"竞争对手"呢！ | 只要不拆家，猫我都爱！ |
| 分享一些拍立得老公 | 拍立得的老公？是用拍立得拍下老公然后立刻洗出来吗$(￣▽￣)$ | 拍立得老公？那我是不是得先摇一摇，等他显影出来？ | 拍立得老公，不是老公，是拍立得。 |
| 刚刚更新完了游戏打了几把，刚开局还没落地就卡住了直接卡闪退了，后面又卡了几秒，怎么感觉更新后bug越来越多了 | 程序员："今天的你，运气真好~ | 看来你的游戏角色还没学会"瞬间移动"技能，直接卡到异次元去了！ | 我玩的都是大逃杀游戏，不是叫你玩模拟飞行。 |
| 从一年级到现在不知道玩了几天了 | 你至少幼儿园没好好玩 | 你这是在练习"游戏学位"吧，快要毕业成"游戏博士"了！ | 6000天 |
| 年轻不嗨，老年痴呆。 | 年轻不浪，老年空想；年轻不闯，老来悲伤。 | 年轻不嗨，老年痴呆；所以现在嗨，老了还能当DJ！ | 老年痴呆，年轻嗨。 |
| 长大是什么感觉，大概是既没成为想成为的人，又丢了原来的自己，既没得到自己一心想要的东西，又失去了原来拥有的快乐。 | 唯一不变的，大概就是一直在变吧！ | 长大是什么感觉？就像升级打怪，结果发现自己选了个最难的副本，还没存档！ | 长大了，就是突然发现自己不是个小孩了。 |
| 持续单身，偶尔心动，惯性拒绝，保持清醒长期快乐，祝我幸福，祝我发财。 | 间歇性渴望爱情，持续性懒得经营。 | 这简直是单身贵族的修炼手册啊！ | 我应该在那个瞬间选择相信你，而不是现在这个样子。 |
| 今天见朋友娃娃，你们猜怎么着？ | 娃娃抓了你？ | 今天见朋友娃娃，你们猜怎么着？——他抓到了一只"空气"！ | 我朋友抓了一只猫。 |
| 大家经历过最恐怖的事情是什么 | 上周所发现没纸了 | 半夜肚子饿，冰箱只有一颗生菜！ | 我刚刚在电梯里被偷了钱包。 |
| 可以帮我想个小说书名吗？ | 《催更了你还想跑》 | 《我在沙发上拯救世界》 | 《我的朋友都是大坏蛋》 |
| 晴天阴天下雨天，你最喜欢哪一天 | 法定节假日 | 我最喜欢放假那一天！ | 我最喜欢晴天，因为晴天的时候，我可以看到太阳。 |
| 请问大家放假很无聊都在做什么？ | 上班 | 放假很无聊？我在练习"葛优躺"大赛，争取拿个金牌！ | 我无聊的时候，就会在脑中想出一个无聊的点子，然后就真的无聊了。 |
| 求小说好句！格式："神说，..........." | 神说，我有点忙，你先跟他们聊着... | 神说，别担心，WiFi密码是12345678。 | "神说，你们这群人类太吵了，都给我闭嘴！" |
| 暑假大家一般几点起床？ | 每天被妈妈骂醒来。 | 下午好！ | 暑假起床时间：12:00 |
| 一个很懂你但他不爱你的人和一个对你有诚挚的爱意你不太懂" | 前者是读者，后者是作者。 | 一个很懂你但他不爱你的人和一个对你有诚挚的爱意但你不太懂——那就找个既懂我又爱我的，毕竟我也不想难为自己！ | 还是别了吧。 |
| 蚂蚁从20米高的楼落下不会死，那我蚂蚁背着我落下会不会死呢？ | 会，因为蚂蚁不会接住你 | 蚂蚁：我选择辞职！ | 蚂蚁会死。 |
| 我歌月徘徊，我摆日更悠。 | 我舞影零乱，我崩半年休。 | 我躺床上刷手机，快乐无边际！ | 我走夜路更悠。 |
| "No one but you." 你又会如何翻译这句话呢？ | 就，只有你啊。 | 除了你，谁都不行，连WiFi都不行！ | "除了你，没有人。" |

Figure 15: Chinese Showcases.

**Failure showcase and Analysis.** The level of creativity is uneven, and the creativity demonstrated in the samples of the training dataset varies significantly. For example, in Case 1, LoL's response is an internet meme, while the better-performing one uses a Chinese proverb for comparison. In terms

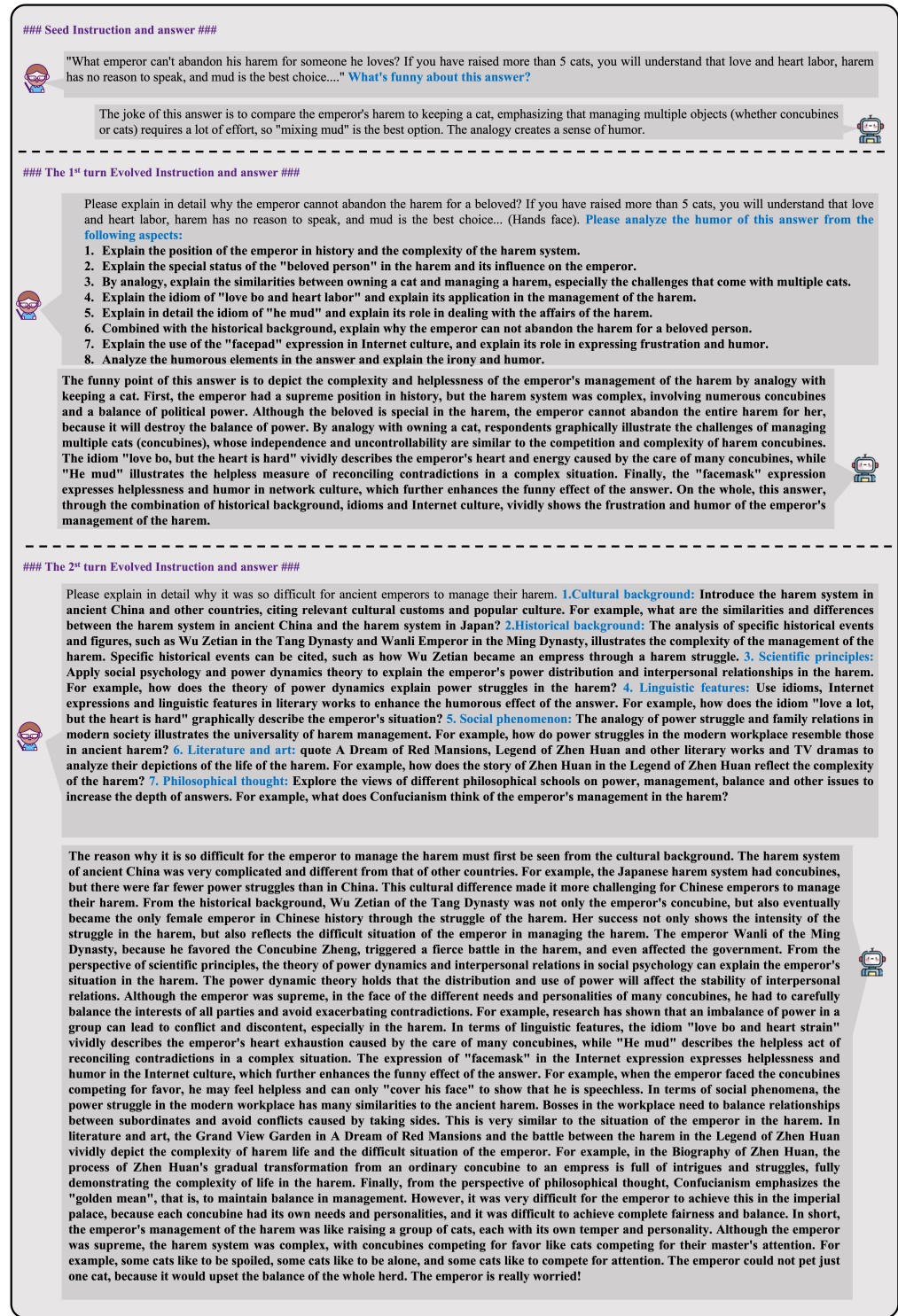

Figure 10: An example of the instruction evolution process in AIE. It stops according to Criterion-1 in Figure 3(b).

of creativity, the latter has an edge. In Case 2, a shorter and antithetical answer is more ingenious. So, There are no failed examples and it's just that according to some people's preferences, they are not humorous.

Table 7: Failure case based on user votes

| Question | failure case | better than LoL (from human) |
|---|---|---|
| What's your relationship like now? | Had a child. | We are like the weather forecast, sometimes sunny, sometimes rainy, forecasts can never keep up with changes. |
| Why does knowing too much make one an orphan of the world? | Because smart people always find it difficult to find companions, either to become rivals, or to be regarded as rivals. | Because knowing too much makes the world too small. |

## A.6 THE DETAILS OF USER STUDY

We conduct a human evaluation to validate LoL's performance in humor generation. We select the first 200 samples from the validation subset of the Ruozhiba dataset* and use the aforementioned method to transform the queries into question-answer pairs. Then, four large language models (LLMs) generate responses to each question, which act as four options. Subsequently, we conduct a user-preference study to directly evaluate the creativity of the LLMs. We present a question and several corresponding replies and ask users to choose the most creative and humorous response. We select four advanced LLMs to generate responses for a total of 200 questions, and the four responses from the four different LLMs are randomly arranged among the options. We conduct an extensive survey through an online survey platform*, ultimately collecting 15 valid questionnaires with 3000 votes. From these collected questionnaires, we calculate the proportion of times each LLM is selected for each question. Finally, we sum up the total number of times each LLM is chosen across all validation samples, as shown in Figure 6(c). The ratio of this sum to the total number of selections among all LLMs represents the user preference for each LLM. Additionally, we calculate the win - rate based on the question dimension, as depicted in Figure 6(b).

### Backdoor criterion

**Q: Why was the cookie sad?**

**A: Because his mom was a wafer long!**

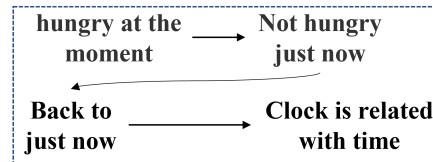

### Frontdoor criterion

**Q: What does a clock do when it's hungry?**

**A: It goes back for seconds.**

Figure 17: Backdoor and Frontdoor criterion examples of humor generation.

---

*https://github.com/Leymore/ruozhiba/tree/main?tab=readme-ov-file
*https://www.wjx.cn/

---

**Algorithm 1** Algorithm of Automatic Instruction Expansion

---
**Require:** Dialogue dataset $D$, initial instruction $I_0$, evolution rules $R_0$
1:  Sample initial dialogue $d_0 = (q, a)$ from $D$
2:  Initialize evolution round counter $t \leftarrow 0$
3:  Set current dialogue $d_{cur} \leftarrow d_0$
4:  Set current instruction $I_{cur} \leftarrow I_0$
5:  Set current rule $R_{cur} \leftarrow R_0$
6:  **while** $t < 3$ **do**
7:     // Instruction Evolution Phase
8:     $I_{new} \leftarrow \text{Generator}(d_{cur}, I_{cur}, R_{cur})$            **// as shown in Figure 3(a)**
9:     $d_{imag} \leftarrow \text{Imaginator}(I_{new}, a)$                   **// as shown in Figure 3(a)**
10:     // Analysis Phase                                **// as shown in Figure 3(b)**
11:     **if** Criterion-2$(d_{imag})$ **then**
12:         **break**
13:     **end if**
14:     **if** $\neg$Criterion-1$(I_{new}, I_{cur})$ **then**
15:         $d_{cur} \leftarrow d_{imag}$
16:         $R_{cur} \leftarrow \text{Analyst}(R_0, I_{cur})$
17:         $t \leftarrow t + 1$
18:     **else**
19:         $I_{cur} \leftarrow I_{new}$
20:         **break**
21:     **end if**
22:  **end while**
23:  Store $I_{cur}, d_{cur}$

---

**Algorithm 2** Algorithm of GESIT

---
**Input:** Policy model $\pi$, Discriminator $\pi^*$, Preference dataset $D_0 = \{(q_i, a_i^+, a_i^-)\}$, Expert $E$.
**Output:** Finetuned policy model $\pi$
1:  $\widetilde{D} \leftarrow \{\}, D_k \leftarrow D_0, k \leftarrow 0,$
2:  **for** each $t \in N$ **do**
3:     sample $(q, a^+, a^-) \in D_k$
4:     $\pi \leftarrow \pi + \nabla L_{DPO}(\pi, (\widetilde{q}, a^+, a^-))$            // DPO training on origin dataset.
5:     **if** $t = T$ **then**
6:         sample $\widetilde{q} \in D_k$                    // Randomly sample questions form origin dataset.
7:         $\widetilde{a}_1, \widetilde{a}_2 \leftarrow \pi(\widetilde{q}), \pi(\widetilde{q})$            // $\pi$ outputs new responses to sampled questions.
8:         $\widetilde{a}^+, \widetilde{a}^- \leftarrow \pi^*(\widetilde{q}, \widetilde{a}_1, \widetilde{a}_2)$   // $\pi^*$ judges both responses as positive and negative samples.
9:         $\widetilde{r}^+, \widetilde{r}^- \leftarrow E(\widetilde{q}, \widetilde{a}^+), E(q, \widetilde{a}^-)$       // $E$ outputs rationales for responses.
10:       $\widetilde{D} \leftarrow \{\widetilde{q}, \widetilde{a}^+, \widetilde{a}^-, \widetilde{r}^+, \widetilde{r}^-\}$
11:       $D_{k+1} \leftarrow D_k \cup \widetilde{D}$             // New data are added into dataset to train.
12:     **end if**
13:  **end for**

---

**Algorithm 3** Inference Step of LoL

---
**Input:** Questions $Q$, LoL-trained LLM $\pi'$, response number $n$, different prompts $P$.
**Output:** The most funny response $A_{best}$
    Select $n$ Prompts $P_{sub}$ from $P$
    $\{A_i\}_{i=0}^n \leftarrow \pi'(Q, P_{sub})$            // $\pi'$ output candidate responses through $P_{sub}$.
    $A_{best} \leftarrow \pi'(Q, \{A_i\}_{i=0}^n)$     // $\pi'$ as judgement model to select the most funny answer.

---

### Seed rule ###

You are an instruction rewriter and need to rewrite the given #Instruction #into a more complex version. Please follow the steps below to rewrite the given #Instruction #into a more complex version. Note that our #Instruction #end goal is to help the model understand the hilarious elements.

**Step 1:** Please read #Instruction #carefully, analyze every word and phrase proposed in #Instruction #, list all the knowledge that may extend to make the instruction more complex (making it more difficult to handle for well-known AI assistants like ChatGPT and GPT-4), which can help the AI assistant understand the punchlines in the conversation as well as deeper background principles. Please do not change the content after the question: and answer: in #Instruction #, only increase the complexity of the prompt! Please do not provide a way to change the language of the instruction!

**Step 2:** Please develop a detailed plan based on the #Methods List #generated in Step 1 to make #Instruction #more complex. The plan should include the methods in #Methods List #.

**Step 3:** Please follow the plan step by step and provide #Rewritten Instruction #that does not change the content of the chat.

**Step 4:** Please review #Rewritten Instruction #carefully and identify any unreasonable parts. Make sure #Rewritten Instruction #is just a more complex version of #Instruction #. Just provide #Finally Rewritten Instruction #without any explanation.

**Step 5:** Please read the #Rewritten Instruction #carefully and write an answer based on the #Rewritten Instruction #.

- - - - - - - - - - - - - - - - - - - - - - - - - - - - - - - - - - - - - - - - - - - - - - - -

### Improved rule ###

You are an instruction rewriter and need to rewrite the given #Instruction# into a more complex version. Please follow the following steps to rewrite the given #Instruction# into a more complex version. Note that the ultimate purpose of our #Instruction# is to help the model understand humorous elements.

**Step 1:** Please read #Instruction# carefully and analyze each word in #Instruction# **word by word**. List all possible knowledge points **and related background information** that can be extended to make the instruction more complex (making it more difficult for well-known AI assistants such as ChatGPT and GPT-4 to handle). **When listing knowledge points, please comprehensively consider factors such as historical background, social culture, psychological theory, literary works**, **political background, religious belief, and other aspects** to help the AI assistant fully understand the joke in the conversation and the deeper background principles. Please do not change the content after question: and answer: in #Instruction#, only increase the complexity of the prompt! Please do not provide methods for changing the instruction language!

**Step 2:** Please formulate a detailed plan according to the #Methods List# generated in Step 1 to make #Instruction# more complex. The plan should include the methods in #Methods List# and **explain in detail how each method is implemented specifically. Ensure that the plan covers multidisciplinary knowledge points and takes into account understanding differences in different cultural backgrounds. The plan should include specific operation steps, expected effects, and possible challenges and solutions.**

**Step 3:** Please implement the plan step by step and provide #Rewritten Instruction#, which does not change the chat content. **The implementation of each step should include specific operation details and expected effects to ensure the complexity and specificity of #Rewritten Instruction#. During the implementation process, record the implementation process and results of each step for subsequent review and improvement.**

**Step 4:** Please carefully review #Rewritten Instruction# and identify any unreasonable parts. Ensure that #Rewritten Instruction# is only a more complex version of #Instruction#. **During the review process, consider the logic, consistency, and operability of the instruction. The review should include a detailed inspection of each step to ensure its rationality and feasibility.** Only provide #Finally Rewritten Instruction# without any explanation.

**Step 5:** Please read #Rewritten Instruction# carefully and write an answer according to #Rewritten Instruction#. **The answer should show the understanding and implementation effect of the complex instruction and reflect the comprehensive application of multidisciplinary knowledge points. The answer should include detailed explanations and application examples of each knowledge point to show the comprehensive understanding and implementation ability of the instruction**.

Figure 18: The seed rule and an example of evolved rule in AIE mentioned in Section 2.2. What is marked in blue is the result after the first rule improvement (line 16 in Algorithm 1), while the text in red is the result after the second rule improvement.

### Seed Instruction and answer ###

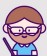

"At what point did you feel you should leave your job? Of course, it's due to physical reasons: my stomach isn't good, the pancake is too big, I can't eat it. My back isn't good, the pot is too heavy, I can't carry it." **What's funny about this answer?**

The joke of this answer is the use of metaphors and puns. Respondents used "bad stomach, cake is too big, can not eat" and "bad waist, pot is too heavy, can not move" to express the image of work pressure, heavy tasks, their own unbearable situation.Comparing work tasks to "pancakes" and "pans", graphically indicating that the task is too heavy.On the surface, it is a physical health problem (bad stomach, bad waist), in fact, it is to express the frustration of work pressure and burden.

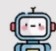

### The 1ˢᵗ turn Evolved Instruction and answer ###

Please explain in detail why you have been feeling under the weather lately, especially stomach problems and low back pain. **Please answer with the following information:** 1. Cultural background: Reference to work culture and stress management methods of different countries and regions, combined with humor elements in popular culture. 2. Historical background: References to famous workaholics or historical figures with health problems caused by work stress. 3. Scientific principles: Explain the effects of stress on the body, especially on the stomach and waist, citing medical research. 4. Linguistic features: Use idioms, proverbs and humorous expressions to enhance the interest and cultural depth of the answers. 5. Other relevant in-depth information: Combining psychological and sociological theories to explain the impact of work stress on psychological and social relationships.

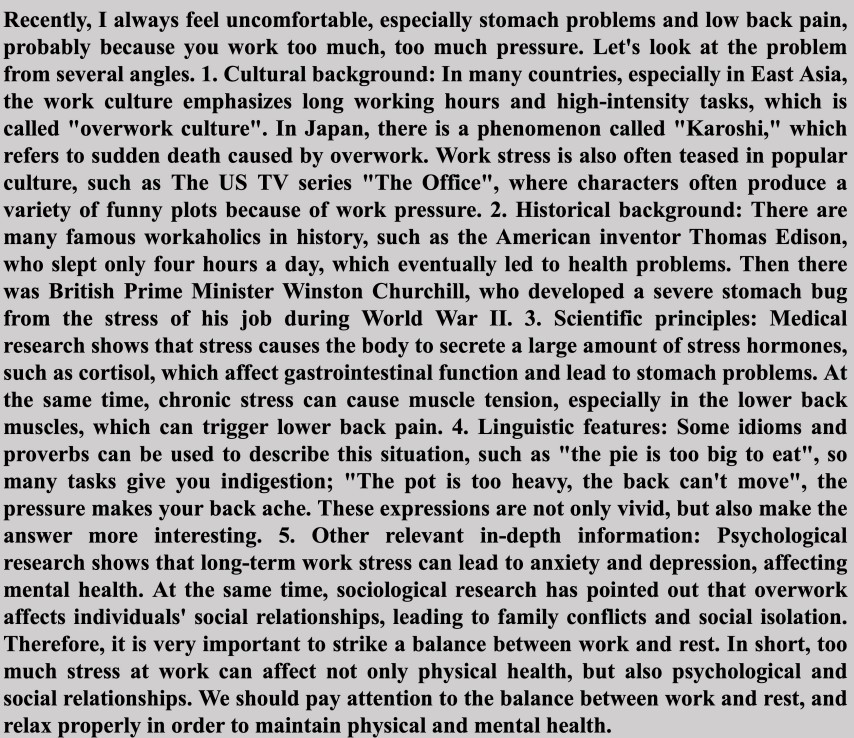

**Recently, I always feel uncomfortable, especially stomach problems and low back pain, probably because you work too much, too much pressure. Let's look at the problem from several angles. 1. Cultural background: In many countries, especially in East Asia, the work culture emphasizes long working hours and high-intensity tasks, which is called "overwork culture". In Japan, there is a phenomenon called "Karoshi," which refers to sudden death caused by overwork. Work stress is also often teased in popular culture, such as The US TV series "The Office", where characters often produce a variety of funny plots because of work pressure. 2. Historical background: There are many famous workaholics in history, such as the American inventor Thomas Edison, who slept only four hours a day, which eventually led to health problems. Then there was British Prime Minister Winston Churchill, who developed a severe stomach bug from the stress of his job during World War II. 3. Scientific principles: Medical research shows that stress causes the body to secrete a large amount of stress hormones, such as cortisol, which affect gastrointestinal function and lead to stomach problems. At the same time, chronic stress can cause muscle tension, especially in the lower back muscles, which can trigger lower back pain. 4. Linguistic features: Some idioms and proverbs can be used to describe this situation, such as "the pie is too big to eat", so many tasks give you indigestion; "The pot is too heavy, the back can't move", the pressure makes your back ache. These expressions are not only vivid, but also make the answer more interesting. 5. Other relevant in-depth information: Psychological research shows that long-term work stress can lead to anxiety and depression, affecting mental health. At the same time, sociological research has pointed out that overwork affects individuals' social relationships, leading to family conflicts and social isolation. Therefore, it is very important to strike a balance between work and rest. In short, too much stress at work can affect not only physical health, but also psychological and social relationships. We should pay attention to the balance between work and rest, and relax properly in order to maintain physical and mental health.**

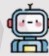

Figure 11: An example of the instruction evolution process in AIE. It stops according to Criterion-2 in Figure 3(b).

### Conversation ###

Question: How do you think about Russian flight?

Answer: I took it once before. It arrived an hour ahead of schedule. Now I have been divorced for three years.

- - - - - - - - - - - - - - - - - - - - - - - - - - - - - - - - - - - - - - - - - - - - - - - - - - - -

### Instruction ###

Please summarize the key words in the question and answer around the core points of the conversation. And please describe the correspondence and causal relationship between the key words in the question and the key words in the answer, as well as the causal relationship between the key words in the answer. If there is no direct causal relationship between the two keywords, please add the full story in the middle to make it logically coherent and have a causal relationship.

Please output strictly in the following format: Keywords: ; The causal relationship between [the second keyword] and [the first keyword] is: ;...

- - - - - - - - - - - - - - - - - - - - - - - - - - - - - - - - - - - - - - - - - - - - - - - - - - - -

### Answer ###

**Keywords:** Shandong Airlines; took it once; arrived an hour ahead of schedule; divorced for three years.

**The causal relationship between [took it once] and [Shandong Airlines] is**: Because the passenger chose Shandong Airlines, so he took a flight of Shandong Airlines once.

**The causal relationship between [arrived an hour ahead of schedule] and [took it once] is**: Because the passenger took a flight of Shandong Airlines once, so he experienced the situation that the flight arrived an hour ahead of schedule.

**The causal relationship between [divorced for three years] and [arrived an hour ahead of schedule] is**: Because the flight arrived an hour ahead of schedule, the passenger returned home early and accidentally discovered the infidelity of his spouse, which led to the breakdown of marriage and finally being divorced for three years.

Figure 12: A showcase of the rationale training in GESIT.

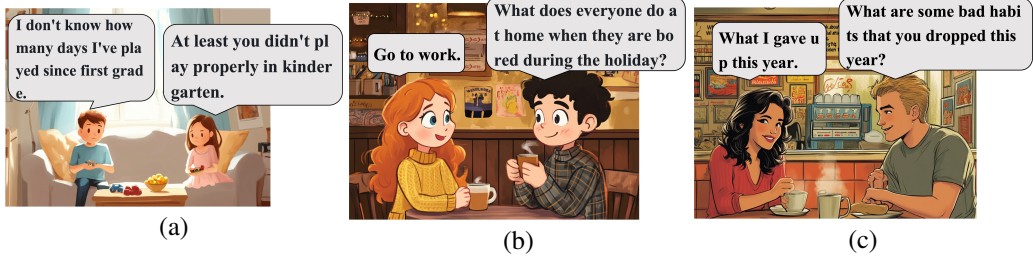

Figure 13: Three showcase of the generation of LoL in English.

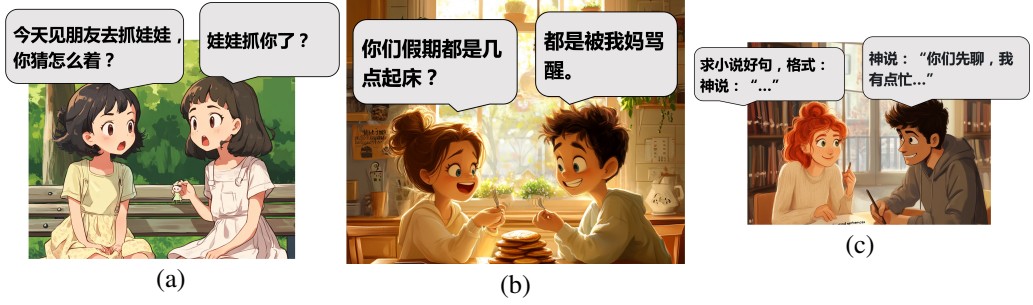

Figure 14: Three showcase of the generation of LoL in Chinese.

| Question | LoL | GPT-4o | CLoT |
|---|---|---|---|
| Am I wrong to be satisfied with my job? | **Job? No no.** | "Of course not! If you're satisfied with your job, you're already one step ahead of the person who invented the snooze button!" | No, you're not wrong to be satisfied with your job. In fact, you're doing just fine. |
| What are some bad habits that you dropped this year? | **What I gave up this year.** | "I gave up my bad habit of hitting the snooze button... Now I just set 10 different alarms instead!" | I stopped asking people to explain their bad habits. |
| I can not help the fact that you look like what you have been through! | **"Looks like you've been through a lot too!"** | "Well, if I look like what I've been through, then I must be a mix of coffee spills, awkward Zoom calls, and a few too many midnight snacks!" | I can't help the fact that you look like what you have been through! |
| What's an insignificant thing that always makes you angry? | **The fact that I can't get stars for being handsome.** | "When people leave just one second on the microwave timer. It's like they enjoy living on the edge of chaos!" | A person who doesn't say "excuse me" when they bump into you. |
| What is a recurring dream you have had? | **"Last night's TV today."** | "I keep dreaming that I'm back in school, but this time I'm the teacher and the students are all my old bosses. Sweet, sweet revenge!" | I have a recurring dream where I'm at a party and I don't know anyone. |
| What's the one thing you can't do without, no matter the price? | **What would you say if someone asked you what you couldn't buy even if you had all the money in the world?** | My morning coffee. Without it, I'm basically a sleep walking zombie with a to-do list!" | A sense of humor. |
| I haven't had a conversation like this since high school | **What were you doing in middle school, then? "** | High school? You mean back when our biggest worry was who stole our lunch money and not how to adult properly?"since high school | I haven't had a conversation like this since high school |
| What do you call a boomer that does not get it? | **What do you call a baby boomer who doesn't understand the situation?** | "A boomerang-they keep coming back with the same questions!" | A boomer who doesn't get it is called a "boomer who doesn't get it." |

Figure 16: English Showcases.

