# OpenReview forum: "Innovative Thinking, Infinite Humor: Humor Research of Large Language Models through Structured Thought Leaps"
_ICLR.cc/2025/Conference — ICLR 2025 Poster_

### Official Review · Reviewer_SKMr · 2024-10-16

**Soundness:** 2
**Presentation:** 2
**Contribution:** 2
**Rating:** 5
**Confidence:** 4

**Summary:**

The paper introduces a framework called Creative Leap of Structured Thought (CLoST) to reinforce humor generation in LLMs. Building upon the limitations of the CLoT paradigm, the authors propose a systematic method inspired by KGs and causal relationships. The framework consists of two stages: Associative Automatic Instruction Evolution (AAIE) and Guided Explorative Self-Improvement Tuning (GESIT). In the first, the model is trained with human-designed instructions to improve its humor-judgement capabilities. In GESIT, the model refines its ability to generate humorous responses through RL, learning from both an expert model and its own judgements. Experiments conducted on English and Chinese humor datasets seemingly demonstrate that CLoST outperforms existing models in humor discrimination and enhances the model's divergent thinking abilities.

**Strengths:**

The paper addresses humor generation, a task that is famously challenging, introducing a framework that uses causal relationships to model associations between concepts. Furthermore, a two-stage training process is a good approach, separating the judgement part from the generation part in LLMs.

**Weaknesses:**

Firstly, the paper lacks clarity in several sections, making it difficult to fully comprehend the proposed methods. For instance, the mathematical formulation in S2.1 seems disconnected from the rest of the paper, and is not well-explained. Secondly, there is insufficient detail about the datasets used, especially the "in-house data". The lack of information about data sources and availability (and code!) raises concerns about the reproducibility of the experiments. Then, the evaluation metrics and experimental setup are not discussed enough. It is unclear in my mind how the multiple-choice questions are constructed and whether the comparisons with baseline models are fair (e.g., are other models trained on the same datasets? If not, than it's a problem).
I will state here, since it's a problem of many papers in this field: the reliance on GPT-4o as an expert model is problematic, as it is a proprietary system. This raises issues regarding the accessibility and reproducibility of your method. Proprietary system's backends change with time and without alerting the users: they are not fit for experiments that need to keep reproducibility to the forefront.
Furthermore, let humans evaluate the generated humor. A good sample of annotators evaluating whether the produced content it is "funny" or not would be a better fit and a real contribution to the field.
Finally, the paper does not address ethical considerations related to HG, such as avoiding offensive or culturally insensitive content.

**Questions:**

- Can you provide more information about the "in-house data" used for training and testing? What are its sources, and is it possible to make this data publicly available to ensure reproducibility?
- How are the multiple-choice questions in the evaluation constructed? Are they standardized across all models, and how do you control for randomness in the selection of options?

---

> ### Author Response · Authors · 2024-11-23
>
> Because the field of jokes is quite new, we appreciate your feedback so that we can better express our innovation.
>
> 1. Not well explained (like Problem definition).
>
> **Humor reasoning is a multi-hop process, and each hop is based on external knowledge injection and proper rationales. Without them, it is difficult for the model to understand the internal humorous logic, making it prone to pattern recognition.** Therefore, **AAIE is proposed to inject and augment knowledge into the original training data. It will be facilitative for LLMs to understand the underlying logic and rationale.**
> And then **in the stage of GESIT, the reasoning logic for each online generated response is extracted** using GPT-4o. In this process, **external knowledge is introduced again to assist the model in logic reasoning and human preferences learning.**
> Experimental results demonstrate that the combination of these two methods can both enhance the model's judgment ability and improve its generative capability.
>
> For example, there is a conversation, “What do you think of Aeroflot?” “I was on this airline once, and arrived an hour early, and finally I have been divorced three years.”. Aeroflot has nothing to do with divorce seemingly. But they could be related by this way, "Aeroflot $\rightarrow$ Well-known feature is fast $\rightarrow$ It could arrive an hour early $\rightarrow$ It leads to trip disrupted $\rightarrow$ So something to ruin the marriage was found $\rightarrow$ Finally the responder is divorced." From the example, we can conclude that there is a sparse knowledge graph between question and answer. **The key of making a creative leap is to mitigate the information insufficiency issue. To realize it, the injection of knowledge beyond the literal is necessary to reasoning, which is the goal of CLoST.**
>
> 2. Datasets details.
>
> The in-house data is private and is not suitable for disclosure. But We have collected Ruozhiba dataset and generate responses by CLoST. They are manually reviewed and screened to be public as research tool. And then we will publish the full code after the paper is accepted.
>
> 3. The evaluation metrics and experimental setup.
>
>  With respect to the multiple-choices questions, there are varying difficulty levels in the dataset, and it is variation primarily reflected in the choice options.
> We collect data from varied humor generation games such as Origi-GO. In these games, a question gains many responses from human and these responses are ranking by human voting. We select response with different votes numbers to construct choices. Easy case is formed by responses with large disparity. Hard case is formed by small difference votes' responses. And with develop of options' number, the gap between options will shrink. Finally, we clarify that we randomly shuffle the options in training and validate sets.

---

> ### Author Response · Authors · 2024-11-23
>
> 4. Experiments.
>
> We appreciate your concerns about fairness, but as an admittedly best LLM, GPT-4o is inevitably used as a baseline for comparison.
> To allay your concerns, we supplemented an experiment on the Ruozhiba dataset, which most well-known LLMs have been trained on. We asked GPT-4o to rewrite the Ruozhiba query into a question and answer pair, placing the punchline in the answer section. Then, we asked GPT-4o again to rewrite the ground-truth answer into a non-humorous version. Based on the positive-negative pair data, LLMs were tested, and the results are shown in Table 9 (we modify it in line 1034 of revised paper). **The results show that CLoST also realizes state-of-the-art performance on Ruozhiba dataset.**
>
> We conduct a human evaluation to validate CLoST's performance in humor generation (\textcolor{blue}{line 430}).
> We choose the first 200 samples in the validation split of the Ruozhiba dataset\footnote{https://github.com/Leymore/ruozhiba/tree/main?tab=readme-ov-file} and the method mentioned above is to make the query into questin-answer pair. Then four LLMs generate responses to each question as four options.
> Then, we conduct a user preference study to directly verify the creativity of the LLMs. We present question and several corresponding replies, and ask users to choose the most creative and humorous responses. We select four advanced LLMs to generate responses for a total of 200 questions, and the four responses from the four distinct LLMs are randomly permuted in the options. We conduct an extensive survey through an online survey platform\footnote{https://www.wjx.cn/}, ultimately collecting 15 valid questionnaires with 3000 votes. Within these collected questionnaires, we calculate the proportion of times each LLM is selected for each question. Finally, we aggregate the total number of times each LLM is chosen across all validation samples, as shown in Figure 6(c). The ratio of this sum to the overall number of selections among all LLMs signifies the user preference for each LLM. Also we calculate the win rate based on the dimension of the problem as shown in Figure 6(b).
>
> Training pipeline (we modify it in line 783 of revised paper):
> CLoST takes a two-stage training strategy. In the first process (supervised fine-tuning (SFT)), we randomly initialize a LoRA model. And we train the model with single-turn question-answer format data (data from Figure 2(a)(b)(c)) and muti-turn question-answer format data (data from Figure 2(d) and AAIE). In the second process (Direct Preference Learning (DPO)), the first stage model serves as ref model, and it is frozen as judgement model. The tunable model is trained to improve the reasoning generation capability. At the beginning of stage 2, only preference question-answer data without rationale is fed into the tunable model for training.  After several steps, the rationale for each online generated response is extracted using GPT-4o and the preference question-answer data with rationles are mixed into the original dataset. And in each batch, the ratio of 'w' and 'w/o' rationale is $1 : 1$.
>
> Experiments details (we modify it in line 792 of revised paper).:
> Our model is fine-tuned based on QWEN1.5-32B-Chat with fine-tuning method LoRA on 8 A100 GPUs. For the first stage, we train the model on the $95\%$ of dataset mentioned above for 6 epochs with AdamW optimizer and the learning rate of $3e-4$. In the second stage, $5\%$ of the dataset are used to train GESIT for 3 epochs with AdamW optimizer and the learning rate of $2e-4$. The models are tested in the tasks introduced in previous part. And the parameters used in generation are list in Table 5.
>
> This article solves the problem that the answer from most LLMs is not humorous and creative enough. In the future, in order to make it more widely used, we will consider the HG problem and offensive or culturally insensitive content you raised.

---

> ### Author Response · Authors · 2024-11-25
>
> Dear Reviewer SKMr,
>
> I hope this message find you well. We have carefully considered your feedback and have made significant improvements to the manuscript. We truly value your insights, and your expertise has greatly contributed to enhancing the quality of our work. Could you please let us know if the revisions meet your expectations?  As the deadline for discussion nears, we kindly ask if you could review our updated paper. We are eager to address any additional questions that may arise. Thank you for your invaluable support and consideration.
>
> Sincerely,
>
> Authors

---

> > ### Comment · Reviewer_SKMr · 2024-11-27
> > **Review Update**
> >
> > Dear authors, I have updated my score to reflect your response to my concerns. However, I cannot go above 5 since the code was still not disclosed and the dataset will remain private, a matter that is of utmost importance for open and reproducible research.

---

> > > ### Author Response · Authors · 2024-11-29
> > >
> > > Dear Reviewer SKMr,
> > >
> > > Thanks for your positive feedback and attention in creative thinking.
> > >
> > > Due to compliance reasons within our company, the data is currently undergoing meticulous review to identify and remove minor amount of content with copyright issues. We are hopeful and confident that this process will be completed before our paper is published. Humor generation is a trending topic that is attracting increasing attention.  **We are committed to releasing both data and code** to ensure **our work is reproducible**. We sincerely appreciate your suggestions so that we can make great improvements in the revised version. And we have the same expectation as you that this innovative work will serve as a milestone for future work and unleash the potential for LLMs in creative thinking.
> > >
> > > Sincerely,
> > >
> > > Authors

---

> > > ### Author Response · Authors · 2024-11-29
> > >
> > > Dear Reviewer SKMr,
> > >
> > > To make it easier for the reviewers to verify the validity of the method, we are inquiring of AC whether there are any means to anonymously share our model weights and inference evaluation code with the reviewers.
> > >
> > > Sincerely,
> > >
> > > Authors

---

### Official Review · Reviewer_JsCo · 2024-11-02

**Soundness:** 4
**Presentation:** 3
**Contribution:** 3
**Rating:** 6
**Confidence:** 3

**Summary:**

This paper introduces CLoST (Creative Leap of Structured Thought), a novel framework for enhancing humor generation and understanding in large language models (LLMs). The work builds upon the Creative Leap-of-Thought paradigm with two key innovations:
1.Automatic Associative Instruction Evolution (AAIE): A multi-agent system for evolving complex instruction sets
2.Guided Exploratory Self-Improvement Tuning (GESIT): A preference optimization approach incorporating causal reasoning
The paper demonstrates improved performance on both Chinese and English humor benchmarks, including SemEval tasks and Oogiri-GO dataset. The framework shows particular strength in handling complex humor discrimination tasks and generating creative responses.

**Strengths:**

1.The paper proposes a novel integration of structured reasoning with creative creation.
2.The paper proposes an innovative multi-agent instruction evolving system and incorporates the causal inference to humor understanding.
3.The proposed two stage approach shows clear performance improvement through comprehensive empirical evaluation across multiple datasets and languages.

**Weaknesses:**

1.The experimental validation lack of human evaluation for generated humor quality.
2.The three-agent system is not well motivated and can be better presented with motivation, clear example and details.
3.Better exploration and analysis of failure cases

**Questions:**

1.The Associative Automatic Instruction Evolution part is a bit hard to understand and can be presented with a short and clear example to go through. It’s confused to interpret the symbols.
2.For the AAIE process, what’s the output data format used to train a lora model?
3.For the guided explorative self-improvement tuning, why do we need r+ and r-? do we use them in the fine-tuning?
4.In Table 1, it’s suprised to see gpt 4o performs worse than Llama3 on SemEval 2020 and Oogiri-GO.
5.How stable is the three-agent system? Are there cases where it fails to converge?

---

> ### Author Response · Authors · 2024-11-23
>
> Because the field of jokes is quite new, we appreciate your feedback so that we can better express our innovation.
>
> 1. AAIE's detail like motivation, examples, output, stability .
>
> **Humor reasoning is a multi-hop process, and each hop is based on external knowledge injection and proper rationales. Without them, it is difficult for the model to understand the internal humorous logic, making it prone to pattern recognition.** Therefore, **AAIE is proposed to inject and augment knowledge into the original training data. It will be facilitative for LLMs to understand the underlying logic and rationale.**
> And then **in the stage of GESIT, the reasoning logic for each online generated response is extracted** using GPT-4o. In this process, **external knowledge is introduced again to assist the model in logic reasoning and human preferences learning.**
> Experimental results demonstrate that the combination of these two methods can both enhance the model's judgment ability and improve its generative capability.
>
> For example, there is a conversation, “What do you think of Aeroflot?” “I was on this airline once, and arrived an hour early, and finally I have been divorced three years.”. Aeroflot has nothing to do with divorce seemingly. But they could be related by this way, "Aeroflot $\rightarrow$ Well-known feature is fast $\rightarrow$ It could arrive an hour early $\rightarrow$ It leads to trip disrupted $\rightarrow$ So something to ruin the marriage was found $\rightarrow$ Finally the responder is divorced." From the example, we can conclude that there is a sparse knowledge graph between question and answer. **The key of making a creative leap is to mitigate the information insufficiency issue. To realize it, the injection of knowledge beyond the literal is necessary to reasoning, which is the goal of CLoST.**
>
> The difficulty in AAIE process can be concluded in two points: 1. How to deepen and broaden understanding step by step. 2. How to end the thought process.  For example in Table 10-15, a simple prompt is difficult to stimulate the understanding of the model like row 1. AAIE utilize LLMs' world knowledge to gradually make the generated prompt contain more information as shown in Table 12 row 1 to 3, simulating the process of human gradual in-depth thinking. So the answers from the two prompts are different like Table 13.
> Additionally, imaginator tries to revivification the conversation based on the informative prompt's answer as shown in Table 10 row 1-2. It makes a shift on story line. So it guides the entire system to explore the boundaries of the conversation, which makes the understanding broader. Finally, If the punchline or core idea of imagined conversations is completely deviated from the original, the system will stop injecting new knowledge, or the maximum step limit is reached.
>
> **The output of AAIE consists of multi-turn question-answer pairs** that include the original conversation, the imagined conversation, and question-answer data of each evolution step.
> Further, some concepts were originally not in the context, but through this method, they have been added to the context. Thus it is easier for LLM model to understand a humor's rationale.
>
> 2. For the guided explorative self-improvement tuning, why do we need r+ and r-? do we use them in the fine-tuning?
>
> **The $r^+$ and $r^-$ for online DPO training are helpful to enhance the humorous reasoning path and weaken inhumorous reasoning path.** It is helpful to simulate the human thinking process. Therefore, $r^+$ and $r^-$ is necessary to realize the reasoning process. We give an example of $r$ in Figure 12 in Appendix (we modify it in line 886 of revised paper)..
>
> In addition, we do not use it in the fist stage i.e. the fine-tuning. Because it is more efficient to train model's preference on DPO training stage.
>
>
> 3. Failure showcase and analysis.
>
> There are some comparative data. Column 2 is the response from CLoST and Column 3 is a better answer by human rating (we modify it in line 1016 of revised paper).
>
> The creativity is uneven, and the creativity shown in the samples of training dataset varies widely.
> For example, in case 1, the response from CLoST is an internet meme and the response from better one using a Chinese proverb to compare.
> In terms of creativity, the latter have broader hands. In case 2, a shorter and antithetical answer is more witty.
> There are no failure examples, only in some people's preferences, it is not humorous.

---

> ### Author Response · Authors · 2024-11-23
>
> 4.Human evaluation for generated humor quality.
>
> Thanks a lot for your suggestion.  We choose the first 200 samples in validate split of Ruozhiba dataset\footnote{https://github.com/Leymore/ruozhiba/tree/main?tab=readme-ov-file}. And we call GPT-4o to rewrite the Ruozhiba query into a question and answer pair, placing the punchline in the answer section. Then four LLMs generate responses to each question as four options.
> And Then we conduct a user preference study to directly verify the creativity of LLMs. We present question and several corresponding replies, and ask users to choose the most creative and humorous responses as shown in . Here we select four advanced LLMs to generate responses for a total of 200 questions, and the four responses from four distinct LLMs are randomly permuted in options.
> We conduct an extensive survey through the online survey platform\footnote{https://www.wjx.cn/}, ultimately collecting 15 valid questionnaires with 3000 votes. Within these collected questionnaires, we can calculate the proportion of times each LLM is selected for each question. Finally, we aggregate the total number of times each LLM is chosen across all validate samples as shown in Figure 6(c). The ratio of this sum to the overall number of selections among all LLMs signifies the user preference for each LLM. Also we calculate the win rate based on the dimension of the problem as shown in Figure 6(b). (we modify it in line 430 of revised paper).

---

> ### Author Response · Authors · 2024-11-25
>
> Dear Reviewer JsCo,
>
> I hope this message find you well. We have carefully considered your feedback and have made significant improvements to the manuscript. We truly value your insights, and your expertise has greatly contributed to enhancing the quality of our work. Could you please let us know if the revisions meet your expectations?  As the deadline for discussion nears, we kindly ask if you could review our updated paper. We are eager to address any additional questions that may arise.
> Thank you for your invaluable support and consideration.
>
> Sincerely,
>
> Authors

---

> > ### Author Response · Authors · 2024-11-29
> >
> > Dear Reviewer JsCo,
> >
> > We sincerely appreciate your suggestions so that we can make great improvements in the revised version. And we have the same expectation as you that this innovative work will serve as a milestone for future work and unleash the potential for LLMs in creative thinking.
> >
> > Sincerely,
> >
> > Authors

---

### Official Review · Reviewer_gQyc · 2024-11-03

**Soundness:** 2
**Presentation:** 1
**Contribution:** 1
**Rating:** 3
**Confidence:** 4

**Summary:**

This paper introduces  the Creative Leap of Structured Thought (CLoST) framework, aiming to improve the humor understanding capabilities of LLMs. It consists of two stages: Associative Automatic Instruction Evolution (AAIE) and Guided Exploratory Self-Improvement Tuning (GESIT). In AAIE, human-designed instructions and automated instruction evolution help the model develop humor understanding capabilities. In GESIT, the model’s humor generation is improved using bootstrap Direct Preference Optimization (DPO) training. By learning from expert-provided joke rationales and AAIE-trained judgment skills, the model iteratively refines its humorous responses through guided reasoning.

**Strengths:**

This paper introduces  the Creative Leap of Structured Thought (CLoST) framework, aiming to improve the humor understanding capabilities of LLMs. It consists of two stages: Associative Automatic Instruction Evolution (AAIE) and Guided Exploratory Self-Improvement Tuning (GESIT).

**Weaknesses:**

1. This paper is very difficult to read and understand. Comes with a seemingly fancy title and introductory paragraphs not closely related with the goal.

It claims "humor generative abilities" in the intro and the teaser figure, half of the method is for improving humor judgement ability and the second half is developed for improving generation abilities. But the test tasks are multiple choice questions of SemEval (humor classification/discrimination) and has nothing to do with generation.

The motivation and method contains lots of unclear expressions. CLoT, the method that it is based on, is not explained clearly. As a reader I am so confused with this paper that I have to rely on AI to help me understand this paper better. Even the figure is confusing. For example in Figure 1, how come that a response by CLoST "Job? No No?" is more humorous and interesting than more informative ones such as "Of course not! If you are satisfied with your job, you're already one step ahead of the person who invented the snooze button" by GPT-4o?

2. The authors seems to lack sufficient background knowledge of NLP&humor basic concepts and existing works. Multiple-choice questions is not a faithful title/motivation to "humor generation abilities".

3. Experiments: The compared baselines are zero-shot LLMs such as GPT, QWEN, etc. The only prior work being compared is CLoT which is not discussed in detail and lacks any ablation of the introduced components.

**Questions:**

/

---

> ### Author Response · Authors · 2024-11-23
>
> Because the field of jokes is quite new, we appreciate your feedback so that we can better express our innovation.
>
> 1. The motivation is not clear.
>
> **Humor reasoning is a multi-hop process, and each hop is based on external knowledge injection and proper rationales. Without them, it is difficult for the model to understand the internal humorous logic, making it prone to pattern recognition.** Therefore, **AAIE is proposed to inject and augment knowledge into the original training data. It will be facilitative for LLMs to understand the underlying logic and rationale**.
> And then **in the stage of GESIT, the reasoning logic for each online generated response is extracted** using GPT-4o. In this process, **external knowledge is introduced again to assist the model in logic reasoning and human preferences learning**.
> Experimental results demonstrate that the combination of these two methods can both enhance the model's judgment ability and improve its generative capability.
>
> For example, there is a conversation, “What do you think of Aeroflot?” “I was on this airline once, and arrived an hour early, and finally I have been divorced three years.”. Aeroflot has nothing to do with divorce seemingly. But they could be related by this way, "Aeroflot $\rightarrow$ Well-known feature is fast $\rightarrow$ It could arrive an hour early $\rightarrow$ It leads to trip disrupted $\rightarrow$ So something to ruin the marriage was found $\rightarrow$ Finally the responder is divorced." From the example, we can conclude that there is a sparse knowledge graph between question and answer. **The key of making a creative leap is to mitigate the information insufficiency issue. To realize it, the injection of knowledge beyond the literal is necessary to reasoning, which is the goal of CLoST.**
>
> 2. Judgement model's importance.
>
> **Judgement ability is a fundamental skill of LLMs to further empower their reasoning ability.** Some researches have verified that reward model is important for reasoning methods such as Monte Carlo tree search or the reinforcement strategy learning.
> **Reward model helps to optimize the behavior of large language models by providing feedback so that they produce outputs that are more in line with expectations.** Due to the subjectiveness of humor, a scalar score may include great noise. **Choosing-best-from-multiple-choices is then a fundamental to establish an evaluation metric.**
>
> 3. CLoT is not be claimed clearly, and humor generation performance compare.
>
> CLoT develops two basic ability to facilitate humor understanding and generation, selection skill and ranking skill. The innovation of CLoT is the introduction of nouns as instruction receipt, thereby enhancing generalization.
> However, as mentioned in CLoT, it directly fine-tuning on the given creative data merely amounts to a rigorous fitting of the data. This fitting process only captures the inherent creative patterns within the data, failing to stimulate ”thinking outside the box” for generating novel ideas. Furthermore, creative data is inherently scarce, and relying solely on dataset fitting easily leads to being trapped in local patterns. (we modify it in line 49 of revised paper).
>
> Regarding the humor generation, humor is subjective, and judgements from diversified groups may vary.
> In terms of expression differences, CLoST could be more human-like and short which leaves more room for imagination. From a statistical perspective in Figure 6(b), **human evaluation shows that CLoST gain more preference.**

---

> ### Author Response · Authors · 2024-11-23
>
> 4. Experiments.
>
> (a) Regarding the experiments on generation, we conduct Divergent Association Task to show the model's imagination capability in Figure 5 (we modify it in line 411 of revised paper). In this experiment, we provided specific words and asked the model to generate associations and imaginations, resulting in 10 associated words. We then used these 11 words to calculate the DAT score where semantic distances are computed.  **The results show that CLoST gains the best performance on divergent association thinking.**
> For humor generation, there are some showcases in Appendix (we modify it in line 912 of revised paper). All showcases show that CLoST could be more human-like and short which leaves more room for imagination.
>
> (b)We conduct a human evaluation to validate CLoST's performance in humor generation (we modify it in line 430 of revised paper).
> We choose the first 200 samples in the validation split of the Ruozhiba dataset\footnote{https://github.com/Leymore/ruozhiba/tree/main?tab=readme-ov-file} and the method mentioned above is to make the query into questin-answer pair. Then four LLMs generate responses to each question as four options.
> Then, we conduct a user preference study to directly verify the creativity of the LLMs. We present question and several corresponding replies, and ask users to choose the most creative and humorous responses. We select four advanced LLMs to generate responses for a total of 200 questions, and the four responses from the four distinct LLMs are randomly permuted in the options. We conduct an extensive survey through an online survey platform\footnote{https://www.wjx.cn/}, ultimately collecting 15 valid questionnaires with 3000 votes. Within these collected questionnaires, we calculate the proportion of times each LLM is selected for each question. Finally, we aggregate the total number of times each LLM is chosen across all validation samples, as shown in Figure 6(c). The ratio of this sum to the overall number of selections among all LLMs signifies the user preference for each LLM. Also we calculate the win rate based on the dimension of the problem as shown in Figure 6(b).
>
> (c)  We appreciate your concerns about fairness, but as an admittedly best LLM, GPT-4o is inevitably used as a baseline for comparison.
> To allay your concerns, we supplemented an experiment on the Ruozhiba dataset, which most well-known LLMs have been trained on. We asked GPT-4o to rewrite the Ruozhiba query into a question and answer pair, placing the punchline in the answer section. Then, we asked GPT-4o again to rewrite the ground-truth answer into a non-humorous version. Based on the positive-negative pair data, LLMs were tested, and the results are shown in Table 9. **The results show that CLoST also realizes state-of-the-art performance on Ruozhiba dataset.** (we modify it in line 1034 of revised paper).
>
> (d) We conduct ablation study about judgement performance in Table 3 and 4 (we modify it in line 379and 447 of revised paper). **In Table 3, row 5-6 shows that model trained with AAIE realizes a noticeable improvement on English benchmarks.**  **Row 3-4 in Table 4 also shows the increasing of performance on Chinese benchmark.** And ablation study on Divergent Association Thinking test is conducted in Figure 5(b) (we modify it in line 411 of revised paper). which shows that **AAIE enhance the divergent associate thinking ability.**

---

> ### Author Response · Authors · 2024-11-25
>
> Dear Reviewer gQyc,
>
> I hope this message find you well. We have carefully considered your feedback and have made significant improvements to the manuscript. We truly value your insights, and your expertise has greatly contributed to enhancing the quality of our work. Could you please let us know if the revisions meet your expectations?  As the deadline for discussion nears, we kindly ask if you could review our updated paper. We are eager to address any additional questions that may arise.
> Thank you for your invaluable support and consideration.
>
> Sincerely,
>
> Authors

---

> > ### Author Response · Authors · 2024-12-01
> >
> > Dear Reviewer gQyc,
> >
> > We deeply appreciate the time and effort you have invested in reviewing our paper. We have thoroughly addressed your valuable comments and made the necessary revisions. Could you kindly re-evaluate our manuscript at your earliest convenience? We are more than willing to discuss any remaining concerns you might have.
> >
> > Thank you for your understanding and cooperation.
> >
> > Sincerely,
> >
> > Authors

---

### Official Review · Reviewer_Unwt · 2024-11-05

**Soundness:** 3
**Presentation:** 3
**Contribution:** 3
**Rating:** 6
**Confidence:** 3

**Summary:**

This paper introduces the Creative Leap of Structured Thought (CLoST) framework, which enhances large language models' ability to generate and recognize humor through structured thinking and self-improvement. CLoST introduces a two-stage approach: first, using Associative Automatic Instruction Evolution (AAIE) to diversify and refine humor judgment through complex instructions, and second, employing Guided Explorative Self-Improvement Tuning (GESIT) to strengthen logical reasoning and humor understanding via reinforcement learning. CLoST improves humor judgment and creativity across multiple language benchmarks.

**Strengths:**

- The paper presents the CLoST framework, a novel methodology for generating humor in LLMs. This structured approach integrates knowledge graphs and causal inference, which helps reinforce logical connections between seemingly unrelated concepts, thus facilitating more coherent humor generation. Notably, the research delves deeply into data augmentation techniques, enhancing the model’s ability to generate diverse and contextually relevant humorous responses.
- The authors conduct extensive testing of CLoST across multiple humor benchmarks. These experiments demonstrate that CLoST consistently outperforms existing humor generation models in terms of accuracy and robustness. The model’s performance improvements are particularly pronounced in both English and Chinese language settings.

**Weaknesses:**

- The inherent subjectivity of humor presents a major challenge. The current approach seems to emphasize pattern recognition, where the model identifies humor based on learned patterns rather than genuinely understanding or reasoning through the humor's intricacies. The empirical results, especially on hard humor tasks, underscore this shortcoming.

- The Associative Automatic Instruction Evolution (AAIE) method employs a multi-agent system—comprising a rewriter, imaginator, and analyst—to iteratively evolve instructions. While this approach is novel and showcases creative engineering, it introduces considerable computational overhead. More critically, the ablation study results suggest that the added complexity yields only marginal improvements in performance. Specifically, CLoST shows gains in four out of seven tasks, and even these are not substantial enough to justify the extensive computational resources required.

**Questions:**

1. Is the inclusion of AAIE necessary? The performance improvement does not seem substantial, as shown in Table 3. It would be beneficial to highlight the best-performing models in the table and provide a detailed analysis of possible reasons.
2. The paper mentions that the dataset includes varying difficulty levels. Is this variation primarily reflected in the choice options, such as 2-out-of-1 or 4-out-of-1 selections? It would be helpful to include a more in-depth analysis of how these different difficulty levels affect model performance.
3. There are minor issues, such as the use of incorrect quotation marks in LaTeX. Additionally, providing more details on experimental parameters, like the temperature setting used, would enhance the clarity and reproducibility of the experiments.

---

> ### Author Response · Authors · 2024-11-23
>
> Because the field of jokes is quite new, we appreciate your feedback so that we can better express our innovation.
>
> 1. Question about not breaking the shackles of pattern recognition.
>
>  And it is true that pattern recognition is an unavoidable for most work related to deep neural networks. As for the process of humor generation, human preference is an unavoidable problem, but beyond human preference, humor generation has its reasoning logic. The highlight of our method is finding a way to accomplish such a special reasoning process.
>
> Humor reasoning is a multi-hop process, and each hop is based on external knowledge injection and proper rationales. Without them, it is difficult for the model to understand the internal humorous logic, making it prone to pattern recognition. Therefore, AAIE is proposed to inject and augment knowledge into the original training data. It will be facilitative for LLMs to understand the underlying logic and rationale.
> And then in the stage of GESIT, the reasoning logic for each online generated response is extracted using GPT-4o. In this process, external knowledge is introduced again to assist the model in logic reasoning and human preferences learning.
> Experimental results demonstrate that the combination of these two methods can both enhance the model's judgment ability and improve its generative capability.
>
> For example, there is a conversation, “What do you think of Aeroflot?” “I was on this airline once, and arrived an hour early, and finally I have been divorced three years.”. Aeroflot has nothing to do with divorce seemingly. But they could be related by this way, "Aeroflot $\rightarrow$ Well-known feature is fast $\rightarrow$ It could arrive an hour early $\rightarrow$ It leads to trip disrupted $\rightarrow$ So something to ruin the marriage was found $\rightarrow$ Finally the responder is divorced." From the example, we can conclude that there is a sparse knowledge graph between question and answer. The key of making a creative leap is to mitigate the information insufficiency issue. To realize it, the injection of knowledge beyond the literal is necessary to reasoning, which is the goal of CLoST.
>
> 2. AAIE's necessity, and ablation experiments on AAIE and analysis.
>
> We supplement ablation study in Table 3 and 4 (we modify it in line 379, 447 of revised paper).. Row 5-6 shows that model trained with AAIE realizes a noticeable improvement. Row 1 in Table 3 is the performance of QWEN-1.5-32B. Row 2-4 show the performance of gradually adding method on Oogiri-GO-en dataset. The results show that with the increase of tasks, especially in the teacher-student system, the performance of judgment is improved. AAIE with Oogiri-GO-en only makes the performance decrease little. It may be cause by overfitting to the divergence of thought on Oogiri-GO-en.
> In addition, the DAT test is conducted on ablation study in Figure 5(b), which shows that AAIE enhance the divergent associate thinking ability.
>
>
> 3. Experiments set up on selection and parameters set up.
>
> Thanks for your suggestion. With respect to the multiple-choices questions, there are varying difficulty levels in the dataset, and its variety is primarily reflected in the choice options.
> We collect data from varied humor generation games such as Origi-GO. In these games, a question gains many responses from human and these responses are ranking by human voting. We select response with different votes to construct choices. A simple case consists of responses with voting numbers that differ significantly, while a hard case consists of responses with voting numbers that differ marginally. And with increase of options' number, the votes number difference between options will shrink. Finally, we clarify that we randomly shuffle the options in training and validate sets.
>
> Training pipeline details are supplemented in Appendix in (we modify it in line 783 of revised paper). and Experiments details used in generation are list in Table 9 (we modify it in line 791 of revised paper).
> More parameters used in generation are list in Table 9.
>
> 4. Quotation error.
>
> I am so sorry for the mistakes and I modify them in the revised version.

---

> > ### Comment · Reviewer_Unwt · 2024-11-25
> >
> > Thank you for your reply and the additional experimental results. My concerns have been resolved.

---

### Comment · Area_Chair_gsN1 · 2024-11-24
**Public discussion phase ending soon**

Dear reviewers,

Thank you for your diligent work on the reviews. Currently the paper has very split scores: 3, 3, 6, 6, and the authors have responded to every single one of the reviews.

All reviewers: did the authors' rebuttals and other reviews affect your score? Please respond before the 26 November to let the authors have time to respond if you need any clarification. Thank you!

Your AC

---

> ### Author Response · Authors · 2024-11-29
>
> Dear Area Chair gsN1,
>
> To facilitate the reviewers' experience and verification of the creativity of the model, our weights and inference code are currently ready and prepared to be made public. However, due to the double-blind reason, we prepared an anonymous GitHub repository but encountered file size limit when committing model weights to GitHub. Since there is no anonymous mode on huggingface, which way would you recommend us sharing the materials while following the double blind policy?
>
> Sincerely,
>
> Authors

---

### Author Response · Authors · 2024-12-02

Dear Reviewers,

Since QwQ is trending in community and performing surprisingly well on various reasoning benchmarks, we included QwQ in humor related benchmarks for comparison. The results show that although QwQ indicates its strong ability on math and code, the paradigm for improve general reasoning ability might not generalize well in humor related tasks,  which involves creative thinking and reasoning with a broad span of thought.  In fact we are the first to propose a systematic paradigm that fits creative thinking tasks well, and we thereby call for attention from the whole community to co-build creative thinking AI applications that was previously recognized as a gift for human only.

Besr Regards,

Authors

---

> ### Author Response · Authors · 2024-12-02
>
> 1. Comparison on Chinese benchmarks.
>
> | Model      |easy|hard|
> | :-----------: | :-----------: | :-----------: |
> |GPT-4o | 64.98 | 63.49  |
> |LLAMA3-8B | 50.72 | 57.44|
> |LLAMA3-70B| 59.48 | 61.22|
> |QWEN1.5-7B|54.82 | 51.71 |
> |QWEN1.5-14B|53.45 | 57.41|
> |QWEN1.5-32B|52.71 | 56.27 |
> |QWEN2-7B| 51.99 |  58.17 |
> |QWEN2-57B| 65.91 | 57.03 |
> |QWEN2.5-32B| 61.53 | 60.46|
> |Baichuan2-13B| 50.56 | 53.61|
> |CLoT-7B |52.12 | 34.46 |
> |**QwQ-32B** | **59.75** | **57.04**|
> |**CLoST-32B**|**90.95** | **69.97**|
>
> In the benchmark, a question gains many responses from human and these responses are ranking by human voting. We select response with different votes to construct choices. A simple case consists of responses with voting numbers that differ significantly, while a hard case consists of responses with voting numbers that differ marginally.
> The results show that CLoST still gain the state-of-the-art performance.

---

> ### Author Response · Authors · 2024-12-02
>
> 2. Comparison on Ruozhiba dataset.
>
> | **CLoST** | GPT-4o | QWEN1.5-32B | CLoT | **QwQ** |
> | :-----------: | :-----------: | :-----------: | :-----------: | :-----------: |
> |**95.35**| 76.40 | 68.85 | 50.20 | **90.80** |
>
> we supplemented an experiment on the Ruozhiba dataset, which most well-known LLMs have been trained on. We asked GPT-4o to rewrite the Ruozhiba query into a question and answer pair, placing the punchline in the answer section. Then, we asked GPT-4o again to rewrite the ground-truth answer into a non-humorous version. Based on the positive-negative pair data, LLMs were tested, and the results are shown in Table 9 (we modify it in line 1034 of revised paper). The results show that CLoST also realizes state-of-the-art performance on Ruozhiba dataset.

---

> ### Author Response · Authors · 2024-12-02
>
> 3.  Comparison on English Benchmarks.
>
> | Model      | SemEval2020| Origin-GO-en | SemEval 2021 2T1 |SemEval 2021 2T1(hard) |SemEval 2021 3T1 |SemEval 2021 4T1 |
> | :-----------: | :-----------: | :-----------: | :-----------: | :-----------: | :-----------: | :-----------: |
> |GPT-4o | 55.08 | 85.09| 85.09 | 60.77| 43.71 | 34.63 |
> |LLAMA3-8B| 59.93 | 72.05| 43.85 | 54.23 | 39.81 | 29.57 |
> |LLAMA3-70B| 60.73 | 88.51| 93.60 |58.08|39.81| 31.82 |
> |QWEN1.5-7B| 50.46 | 36.65 | 62.04 | 52.02 | 31.54 | 24.89 |
> |QWEN1.5-14B| 50.38 | 53.73| 82.05 | 51.04 | 30.92 | 24.24 |
> |QWEN1.5-32B| 56.39 | 68.01| 68.01 | 52.57 | 35.38 | 28.79 |
> |QWEN2-7B| 50.08 | 62.11  | 56.55 | 50.63 | 32.31 | 23.38 |
> |QWEN2-57B| 48.29 | 48.14 | 83.30 | 52.02 | 37.08 | 28.79 |
> |QWEN2.5-32B| 58.71 | 81.68 | 94.00 | 55.22 | 34.77 | 27.92 |
> |Baichuan2-13B| 51.45 | 50.00 | 51.70 | 52.71 | 35.69 | 24.24 |
> |CLoT- 7B |53.50 | 52.49|  52.49 | 51.74 | 34.46 | 23.59|
> | **QwQ**    |**56.58**| **59.63**|  **80.05**       |**53.06**       |**33.49**       |**24.66**      |
> | **CLoST**  |**64.57**| **97.20** | **96.58**        | **57.45**       | **48.06**       | **35.90**   |
>
> We collect data from varied humor generation games such as Origi-GO. In these games, a question gains many responses from human and these responses are ranking by human voting. We select response with different votes to construct choices. A simple case consists of responses with voting numbers that differ significantly, while a hard case consists of responses with voting numbers that differ marginally. And with increase of options' number, the votes number difference between options will shrink. Finally, we clarify that we randomly shuffle the options in training and validate sets. CLoST also is the SOTA method on these English benchmarks.

---

### Meta-Review · Area_Chair_gsN1 · 2024-12-22

**Metareview:**

The paper proposes a new framework called Creative Leap of Structured Thought (CLoST) to reinforce humor understanding by LLMs. The authors propose a systematic method inspired by KGs and causal relationships. The framework consists of two stages: Associative Automatic Instruction Evolution (AAIE) with human-designed instructions, and Guided Explorative Self-Improvement Tuning (GESIT) with RL, learning from both an expert model and its own judgements. Experiments conducted on English and Chinese humor datasets seemingly demonstrate that CLoST significantly outperforms existing models in humor discrimination and enhances the model's divergent thinking abilities.

Strengths:
Novel approach to humor generation in LLMs (Unwt, JsCo, SKMr)
Uses structured thinking and knowledge graphs (SKMr, Unwt, JsCo)
Shows improved performance on several humor benchmarks (Unwt, JsCo)

Weaknesses:
Reviewer (gQyc) found the method complex, with unclear explanations.
Reviewer (SKMr) said the paper lacked human evaluation of generated humor (but this was addressed)

Based on the scores (3, 5, 6, 6), the paper is just below the acceptance bar but one reviewer (gQyc) did not engage the authors' rebuttal, and another reviewer (SKMr) made unreasonable requests to release training data and code for review (rather than with the paper). I am proposing that the lower scores could be bumped by a notch, and the paper would meet the acceptance bar.

**Additional Comments On Reviewer Discussion:**

The authors provided extensive rebuttals with an additional experiment. Three reviewers (SKMr, Unwt, JsCo) engaged the authors, reviewer gQyc did not reply to the rebuttals.
Reviewer SKMr lamented the fact that code and weights had not been released in time for the review, but this is not grounds for rejection, and the authors make a valid point about checking training data for copyright.

---

### Decision · Program_Chairs · 2025-01-22

Accept (Poster)